# MixTraining: A Better Trade-Off Between Compute and Performance

**Zexin Li**[*]                                                                      *zli536@ucr.edu*
*University of California, Riverside*

**Jiancheng Zhang**[*]                                                               *jzhan745@ucr.edu*
*University of California, Riverside*

**Yufei Li**                                                                          *yli927@ucr.edu*
*University of California, Riverside*

**Yinglun Zhu**[†]                                                                    *yzhu@ucr.edu*
*University of California, Riverside*

**Cong Liu**                                                                          *congl@ucr.edu*
*University of California, Riverside*

**Reviewed on OpenReview:** <https://openreview.net/forum?id=NVpS2g9KRo>

## Abstract

Integrating self-supervised learning (SSL) prior to supervised learning (SL) is a prevalent strategy for enhancing model performance, especially in scenarios with limited labeled data. Nonetheless, this approach inherently introduces a trade-off between computational efficiency and performance gains. Although SSL significantly improves representation learning, it necessitates an additional and often computationally expensive training phase, posing substantial overhead in resource-constrained environments. To mitigate these limitations, we propose MIXTRAINING, a novel training framework designed to interleave multiple epochs of SSL and SL within a unified *mixtraining phase*. This phase enables a seamless transition between self-supervised and supervised objectives, facilitating enhanced synergy and improved overall accuracy. Additionally, MIXTRAINING consolidates shared computational steps, thereby reducing redundant computations and lowering overall training latency. Comprehensive experimental evaluations demonstrate that MIXTRAINING provides a superior trade-off between computational efficiency and model performance compared to conventional training pipelines. Specifically, on the TinyImageNet dataset using the ViT-Tiny model, MIXTRAINING achieves an absolute accuracy improvement of 8.81% (a relative gain of 18.89%) while concurrently accelerating training by 1.29×.

## 1 Introduction

Self-supervised learning (SSL) has emerged as a powerful paradigm for learning general representations from unlabeled data, significantly improving performance on downstream tasks (Brown et al., 2020; He et al., 2022). The standard approach combines SSL with supervised learning (SL) into a two-phase pipeline: first training a model on unlabeled data to learn general representations, then adapting it to specific tasks using labeled data. This SSL+SL pipeline has become the de facto standard across computer vision (Chen et al., 2020; He et al., 2020, 2022; Wang et al., 2023b), natural language processing (Devlin et al., 2019; Radford

---

[*]Equal contribution
[†]Corresponding author

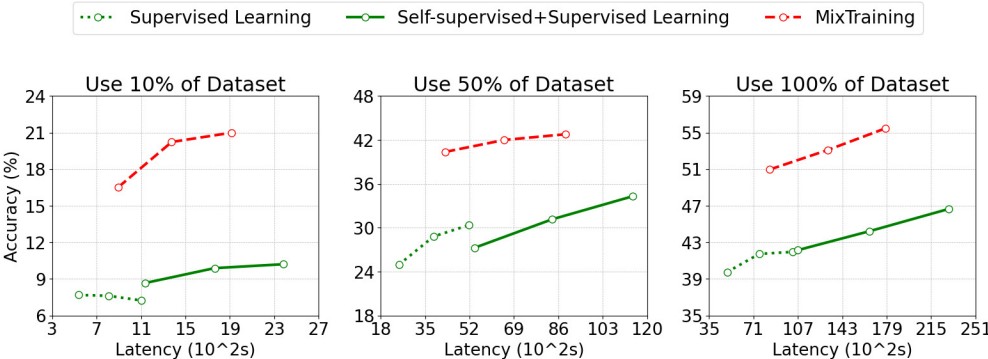

Figure 1: MIXTRAINING demonstrates significant accuracy and computation gains over standard self-supervised learning and supervised learning pipeline across various data limitations levels (10%, 50%, and 100%). Experiments are conducted on the TinyImageNet dataset with the ViT-Tiny model; Each set of three points on the same line represents results obtained with different training durations (50, 75, and 100 epochs).

et al., 2019; Brown et al., 2020; Ouyang et al., 2022; Min et al., 2023), and speech recognition (Hsu et al., 2021; Chen et al., 2022), particularly excelling in data-scarce scenarios where labeled examples are limited.

However, this two-phase pipeline introduces a fundamental trade-off between computational efficiency and model performance. While the first phase of SSL improves downstream accuracy, it requires substantial additional computation that can be prohibitive in resource-constrained environments. In fact, the SSL phase often demands hundreds or thousands of GPU-days before any task-specific training begins (Kaplan et al., 2020; Dosovitskiy et al., 2020; Chung et al., 2024). This computational overhead creates a significant barrier for practitioners, particularly those who must train models from scratch due to data privacy requirements, domain-specific constraints, or the absence of suitable pretrained models.

The abrupt transition between SSL and SL phases in the standard pipeline also presents optimization challenges. The model must suddenly shift from optimizing self-supervised objectives (e.g., reconstruction loss) to supervised objectives (e.g., classification loss), potentially causing training instability and suboptimal adaptation (Peters et al., 2019; Mosbach et al., 2020; Kumar et al., 2022). This discontinuous transition may prevent the model from fully leveraging the synergies between self-supervised and supervised learning signals.

To address these limitations, we propose MIXTRAINING, a novel training framework that introduces a dedicated mixtraining phase between the SSL and SL phases. Our key insight is that the rigid separation between SSL and SL phases in conventional pipelines is neither necessary nor optimal. As illustrated in Fig. 2, the mixtraining phase jointly optimizes both self-supervised and supervised objectives, creating a smooth transition that better preserves learned representations while adapting to downstream tasks. Our design also consolidate what would be separate forward and backward passes in the standard pipeline into unified operations (see Fig. 3 for details), substantially reducing training time while improving model performance.

As shown in Fig. 1, MIXTRAINING consistently achieves superior accuracy and lower training latency across varying data availability levels and training durations. For example, on the full TinyImageNet dataset (Le & Yang, 2015) with reconstruction-based SSL (Hinton & Zemel, 1993; He et al., 2022), MIXTRAINING delivers an 18.89% relative improvement in accuracy (8.81% absolute) and achieves 1.29× speedup over the standard SSL+SL pipeline. The advantage becomes even more pronounced under severe data constraints: at 10% data availability, MIXTRAINING improves accuracy by 105.58% (relative) and reduces training latency by 1.24×.

In this work, we focus on the widely used reconstruction-based self-supervised learning (SSL) paradigm (e.g., MAE) followed by supervised learning (SL) with classification losses. Within this scope, MIXTRAINING consistently provides a better trade-off by smoothing the transition between reconstruction-based SSL and SL while reducing redundant computation. We view extending MIXTRAINING to other SSL objectives (e.g., contrastive methods) as an important direction for future work.

**Contributions.** We introduce the MixTraining framework, which enhances both computational efficiency and model performance over the conventional SSL+SL pipeline. Focusing on reconstruction-based SSL and classification-based SL, our core contributions are:

- **Superior accuracy through smooth objective transition.** MixTraining introduces a mixtraining phase that creates a gradual transition between self-supervised and supervised objectives, avoiding the abrupt shift that can destabilize training in standard pipelines. This smooth transition leads to consistently higher accuracy across all evaluated settings. On TinyImageNet with ViT-Tiny, this design achieves an 18.89% relative accuracy gain over the standard SSL+SL baseline.

- **Computational efficiency via unified forward/backward passes.** By merging SSL and SL epochs into a joint optimization phase, MixTraining consolidates separate forward and backward passes over the backbone into single operations. Since backbone computations typically dominate training cost (the backbone is usually much larger than task-specific heads), this system-level optimization yields substantial speedups (e.g., $1.29\times$ for ViT-T on TinyImageNet) while simultaneously improving accuracy.

- **Robust generalization across diverse settings.** MixTraining demonstrates consistent Pareto improvements over SSL+SL baselines across different experimental configurations, including multiple architectures (ViT, ResNet), datasets (CIFAR-10, CIFAR-100, TinyImageNet), data limitation levels (from 10% to 100%), and training epochs. The framework's modular design enables integration with existing SSL+SL pipelines with minimal modifications.

**Scope and Applicability.** It is important to clarify that MixTraining does not aim to replace the existing pretraining-finetuning paradigm when high-quality pretrained models are readily available. Instead, our framework primarily targets scenarios where practitioners must train models from scratch, such as when data privacy constraints prevent the use of pretrained models, when domain-specific requirements render generic pretrained models inadequate, or when suitable pretrained models are unavailable. In these settings, MixTraining provides a more efficient alternative to the standard SSL+SL pipeline, achieving superior performance while reducing training latency. In Section 5, we provide initial evidence that MixTraining can be effectively applied to pretrained models as a *mid-training* strategy.

**Organization.** The remainder of this paper is structured as follows. Section 2 reviews related work. Section 3 introduces the MixTraining framework, detailing its design intuition and operational procedure. Section 4 presents comprehensive empirical evaluations across multiple datasets and architectures. Section 5 concludes with a discussion of limitations and future directions.

## 2 Related Work

**Learning Pipelines and the Compute-Performance Trade-off.** Early deep learning models primarily relied on supervised learning (SL) with large labeled datasets (Krizhevsky et al., 2012; Simonyan & Zisserman, 2014; He et al., 2016). More recently, self-supervised learning (SSL) has emerged as a powerful paradigm for learning transferable representations without labels, giving rise to a training pipeline that first leverages SSL for representation learning and then adapts the model to downstream tasks via supervised learning. This SSL+SL pipeline has been widely adopted across domains, including computer vision (Chen et al., 2020; He et al., 2020, 2022; Wang et al., 2023b), natural language processing (Devlin et al., 2019; Radford et al., 2019; Brown et al., 2020; Ouyang et al., 2022; Min et al., 2023), and speech recognition (Hsu et al., 2021; Chen et al., 2022). While SSL+SL consistently improves accuracy over SL alone, especially in low-label regimes, it incurs a substantial increase in compute due to the additional SSL phase (Kaplan et al., 2020; Chung et al., 2024), leading to a compute-performance trade-off between the two approaches. Recent work has begun to examine aspects of this trade-off; for example, Liu et al. (2024) explore strategies to reduce the compute cost of the SSL stage. In this work, we directly study on the compute-performance trade-off between SSL+SL and SL, and propose a new MixTraining pipeline that adds a dedicated mixtraining phase between SSL and SL, yielding Pareto improvements in both accuracy and compute efficiency over the standard SSL+SL pipeline.

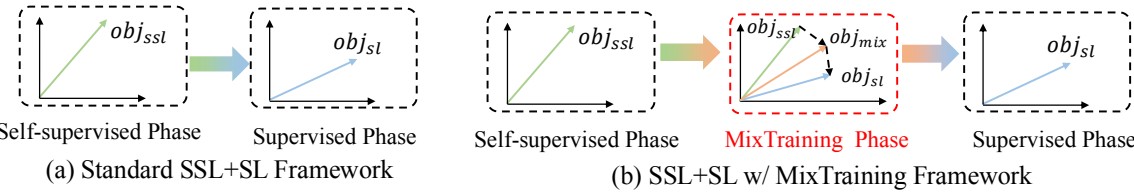

Figure 2: Comparison of MixTraining with the standard SSL+SL framework. (a) The standard SSL+SL framework features an abrupt transition from self-supervised objective ($obj_{ssl}$) to supervised objective ($obj_{sl}$). (b) Our MixTraining framework creates an dedicated mixtraining phase in the middle. The mixtraining phase optimizes towards a mixed objective ($obj_{mix}$), which enables a smooth transition from the self-supervised objective to the supervised objective.

**Modifications to Learning Objectives.** A broad class of methods modifies learning objectives to improve generalization, robustness, or stability. One important direction explores the synergy between different objectives, such as combining self-supervised learning and supervised learning objectives. Prior studies in this space have focused on domain adaptation (Pan et al., 2020; Berthelot et al., 2021), mitigating catastrophic forgetting (He et al., 2021; Mehta et al., 2023), and improving data efficiency (Zhai et al., 2019; Khosla et al., 2020; Yao et al., 2022; Zhang et al., 2024). Another related line of work mixes up data from different sources or domains to bridge distribution gaps or regularize decision boundaries (Zhang et al., 2017; Yun et al., 2019; Verma et al., 2019; Liu et al., 2022; Zou et al., 2023), often by constructing intermediate representations that incorporate information from multiple domains or tasks. Our work is inspired by these lines of research but differs in both purpose and design: we introduce a dedicated mixtraining phase between SSL and SL to smooth the transition between two learning different objectives, and we design this phase for compute efficiency by merging forward and backward passes over the shared backbone—an aspect not addressed in prior work.

**Efficient Training of Deep Neural Networks.** A substantial body of research has focused on reducing the computational and data costs of training deep neural networks while maintaining competitive performance. Model compression techniques (Han et al., 2015a,b; Jacob et al., 2018; Frankle & Carbin, 2018; Blalock et al., 2020; Li et al., 2023) reduce network size and complexity through pruning, weight sharing, quantization, or other architectural simplifications, thereby lowering both memory footprint and FLOPs for more efficient training and deployment. Parameter-efficient finetuning methods (Hu et al., 2021; Ding et al., 2023; Han et al., 2024) freeze most of the backbone and introduce small trainable modules or low-rank adaptations, greatly reducing optimization cost while preserving downstream performance. Data-efficient training strategies (Bengar et al., 2021; Mindermann et al., 2022; Wang et al., 2023a; Bhatt et al., 2024) accelerate convergence by prioritizing informative examples, filtering redundant or noisy data, or leveraging pretrained models for improved initialization. These approaches target model, parameter, or data-level efficiency, and are complementary to our pipeline-level strategy: MixTraining could be integrated with these techniques to further improve its efficiency.

## 3 Methodology

We briefly introduce the standard self-supervised learning and supervised learning (SSL+SL) pipeline in Section 3.1. We introduce our MixTraining framework in Section 3.2, which consists of its design intuition (Section 3.2.1) and operational procedure (Section 3.2.2).

### 3.1 Background: The Standard Self-supervised Learning and Supervised Learning Pipeline

The standard self-supervised learning and supervised learning pipeline consists of two separate phases: a *self-supervised learning (SSL)* phase and a *supervised learning (SL)* phase. In the self-supervised learning phase, a backbone model with a self-supervised learning head is trained to help the model learn general feature representations. Specifically, this process usually relies on learning from unlabeled data with reconstruction

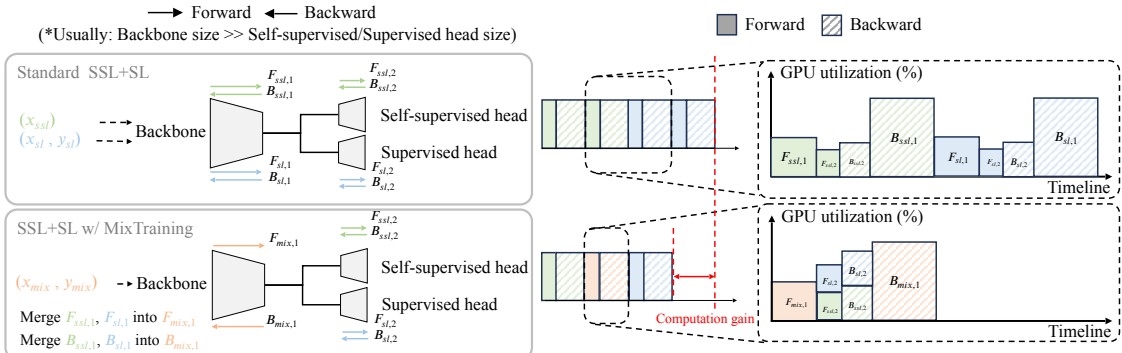

Figure 3: Comparison of MixTraining with the standard SSL+SL framework. MixTraining achieves computation gains over the standard SSL+SL framework. *Top: Standard SSL+SL.* Data first goes through a self-supervised learning pass ($F_{\mathsf{ssl},1} \to F_{\mathsf{ssl},2} \to B_{\mathsf{ssl},2} \to B_{\mathsf{ssl},1}$) and then goes through a supervised learning pass ($F_{\mathsf{sl},1} \to F_{\mathsf{sl},2} \to B_{\mathsf{sl},2} \to B_{\mathsf{sl},1}$). *Bottom: MixTraining.* We merge two forward passes ($F_{\mathsf{ssl},1}$ and $F_{\mathsf{sl},1}$) over the backbone model together into a single pass $F_{\mathsf{mix},1}$, and use its result for both self-supervised head and supervised head; the backward passes ($B_{\mathsf{ssl},1}$ and $B_{\mathsf{sl},1}$) are merged into $B_{\mathsf{mix},1}$ (bottom left). Our modifications reduce computation requirements and allow better parallelization (bottom right).

tasks (Hinton & Zemel, 1993; Hinton & Salakhutdinov, 2006; Michelucci, 2022) or predicting manually masked tokens (Devlin et al., 2019; He et al., 2022). The backbone model is further refined in the supervised learning phase, together with a supervised learning head. This step adapts the model to downstream tasks, usually achieving better performances than directly training the downstream tasks. Fig. 2(a) shows the standard SSL+SL framework, which has now become the go-to approach for improving the performance of AI models (Wang et al., 2023b; Min et al., 2023).

### 3.2   A New Framework: MixTraining

#### 3.2.1   Design Intuition behind MixTraining

While the standard SSL+SL framework has achieved remarkable success, its self-supervised learning and supervised learning phases are completely separated, leaving limited opportunity for interaction or further optimization. To enable closer interactions between these two phases, we propose a novel MixTraining framework—as shown in Fig. 2—which effectively merges several self-supervised learning and supervised learning epochs into an additional *mixtraining phase*, featuring a smooth transition between learning objectives.

MixTraining introduces a new mixtraining phase that allows joint updates of self-supervised and supervised objectives, which is in contrast with the standard SSL+SL framework where one *first* updates the self-supervised objective and *then* updates the supervised objective. At a high level, the benefits of this phase are rooted in the dedicated joint optimization objective. The joint objective can be easily understood as a weighted average of the self-supervised objective and the supervised objective to balance these two objectives. We next explain the design intuition behind the mixtraining phase to achieve both *computation gains* and *accuracy gains*.

**Accuracy Gains.**   Since the self-supervised learning phase aims at learning general representations and the following supervised learning phase aims at learning task-specific information, intuitively, these two phases optimize the model in different directions. The standard SSL+SL framework features an abrupt change in optimization directions during the transition from self-supervised learning and supervised learning (Fig. 2(a)), which may cause instability in model performance (Mosbach et al., 2020). Indeed, there are also studies show that, in certain settings, supervised learning in the second phase can lead to worse model performance (Peters et al., 2019; Kumar et al., 2022). In our MixTraining framework, the mixtraining phase creates a middle ground, i.e., a weighted combination of two objectives, allowing a rather smooth transition from the self-supervised learning objectives to the supervised learning objective, as illustrated in Fig. 2(b). We

hypothesize that such a smooth transition avoids instability in phase transition, thus allowing the model to better adapt to the target task and achieve higher accuracy. We validate the smooth-transition intuition by measuring the cosine similarity of the last-backbone-layer gradients induced by different learning objectives. The average cosine similarity between SSL and SL objectives is only 0.018, indicating that these two objectives are nearly orthogonal. In contrast, the cosine similarity between SSL and MixTraining objectives is 0.791, and between SL and MixTraining objectives is 0.598, suggesting that the MixTraining objective indeed provides a smooth transition from SSL to SL. Our empirical results in Section 4 further support the hypothesis that smooth transitions lead to higher accuracy.

**Computation Gains.** In the standard SSL+SL framework, we first run self-supervised learning passes with data $(x_{\mathsf{ssl}}, y_{\mathsf{ssl}})$, and then run supervised learning passes with data $(x_{\mathsf{sl}}, y_{\mathsf{sl}})$. This process involves forward/backward passes of both $(x_{\mathsf{ssl}}, y_{\mathsf{ssl}})$ and $(x_{\mathsf{sl}}, y_{\mathsf{sl}})$, and strictly follows a *sequential order* to compute each sub-processes (top part of Fig. 3). In contrast, MixTraining aims to jointly optimize self-supervised and supervised objectives. Specifically, it "merges" $(x_{\mathsf{ssl}}, y_{\mathsf{ssl}})$ and $(x_{\mathsf{sl}}, y_{\mathsf{sl}})$ into a *mixed data* $(x_{\mathsf{mix}}, y_{\mathsf{mix}})$ and thus merging the separate forward passes over the backbone model into a single pass, and use its result for both self-supervised head and supervised head; from the computation aspect, we also merge the backward passes of self-supervised learning and supervised learning tasks over the backbone model into a single pass (bottom left part of Fig. 3). Since the size of the backbone model is usually much larger than the size of self-supervised and supervised heads (He et al., 2022; Du et al., 2021; Yang et al., 2023), the merge of forward/backward passes over the backbone model allows us to reduce computation compared to the synchronous setting substantially. Additionally, the merge of forward/backward passes over the backbone model allows better parallelization of forward/backward passes over the self-supervised and supervised heads (right part of Fig. 3), which further speeds up the computation. Empirically, we find the runtime of a single merged mixtraining pass (0.317s) is lower than the combined runtime of an SL pass and an SSL pass (0.543s = 0.248s + 0.295s), validating our intuition.

### 3.2.2 The MixTraining Procedure

In this section, we introduce the operational procedure of our MixTraining framework in detail. Besides the number of self-supervised learning epoch $e_{\mathsf{ssl}}$ and the number of supervised learning epoch $e_{\mathsf{sl}}$, MixTraining takes as input a hyperparameter MIX-RATIO $\rho \in [0, 1]$ to determine the number of self-supervised learning/supervised learning epochs $e_{\mathsf{mix}}$ to be merged into the mixtraining phase. We set

$$e_{\mathsf{mix}} = \lfloor \rho \min(e_{\mathsf{ssl}}, e_{\mathsf{sl}}) \rfloor. \tag{1}$$

Our MixTraining framework then operates by running (i) the vanilla self-supervised learning phase for $e_{\mathsf{ssl}} - e_{\mathsf{mix}}$ epochs, (ii) the mixtraining phase for $e_{\mathsf{mix}}$ epochs, and (iii) the vanilla supervised learning phase for $e_{\mathsf{sl}} - e_{\mathsf{mix}}$ epochs, as shown in Algorithm 1.

Since the self-supervised learning and supervised learning phases are standard, in the following, we mainly discuss the mixtraining phase. In the mixtraining phase, we (i) design a mixing function $g$ to generate *a mixed dataset*, and (ii) design *a joint optimization objective* as supervision signal. We next highlight the design choice for these two parts.

**The Mixed Dataset.** The goal of creating a mixed dataset $\mathcal{D}_{\mathsf{mix}} = g(\mathcal{D}_{\mathsf{ssl}}, \mathcal{D}_{\mathsf{sl}})$ is to extract information stored in self-supervised learning dataset $\mathcal{D}_{\mathsf{ssl}} = \{x_i\}_i$ and supervised learning dataset $\mathcal{D}_{\mathsf{sl}} = \{(x_i, y_i)\}_i$, featuring a smooth transition between two learning objectives Fig. 2 (b). In the simple case where self-supervised learning and supervised learning use the same input (i.e., $x_i$), we can simply set $g(\mathcal{D}_{\mathsf{ssl}}, \mathcal{D}_{\mathsf{sl}}) = \mathcal{D}_{\mathsf{sl}}$. We remark that the importance of this simple case is usually overlooked: conducting self-supervised learning and supervised learning on the same ImageNet dataset allows one to boost the top-1 classification accuracy from 82.5% to 84.9%, without using extra data (He et al., 2022).

We next discuss the general case where the self-supervised learning dataset is not the same as the supervised learning dataset, i.e., $\mathcal{D}_{\mathsf{ssl}} \neq \mathcal{D}_{\mathsf{sl}}$. Inspired by the mixup method in machine learning to improve the generalization and robustness to adversarial examples (Zhang et al., 2017), we consider a *randomized* mixing function $g$, which randomly mixes up data points from both datasets. Specifically, we set

$$\mathcal{D}_{\mathsf{mix}} = g(\mathcal{D}_{\mathsf{ssl}}, \mathcal{D}_{\mathsf{sl}}) = \{(x_{\mathsf{mix}}, y_{\mathsf{sl}}) : x_{\mathsf{mix}} = \lambda\, x_{\mathsf{sl}} + (1 - \lambda)\, x_{\mathsf{ssl}}, (x_{\mathsf{sl}}, y_{\mathsf{sl}}) \in \mathcal{D}_{\mathsf{sl}}, x_{\mathsf{ssl}} \in \mathcal{D}_{\mathsf{ssl}}\}, \tag{2}$$

---

**Algorithm 1** The MixTraining Framework

---

**Input:** Self-supervised learning epoch $e_{\sf ssl}$, Supervised learning epoch $e_{\sf sl}$, MIX-RATIO $\rho \in [0, 1]$.

1: Initialize model parameters $\theta$.
2: Calculate mixtraining epoch $e_{\sf mix}$ via Eq. (1).
3: ▷ Vanilla Self-supervised Learning Phase
4: **for** $e = 1$ to $e_{\sf ssl} - e_{\sf mix}$ **do**
5:     Conduct vanilla self-supervised learning phase *w.r.t.* self-supervised loss $\ell_{\sf ssl}(x; \theta)$ to optimize $\theta$.
6: ▷ MixTraining Phase
7: **for** $e = 1$ to $e_{\sf mix}$ **do**
8:     Optimize the model parameter $\theta$ with respect to the joint optimization objective as in Eq. (3).
9: ▷ Vanilla Supervised Learning Phase
10: **for** $e = 1$ to $e_{\sf sl} - e_{\sf mix}$ **do**
11:     Conduct vanilla supervised learning phase *w.r.t.* supervised loss $\ell_{\sf sl}(f(x), y; \theta)$ to optimize $\theta$.

---

where for each $(x_{\sf sl}, y_{\sf sl}) \in \mathcal{D}_{\sf sl}$, we randomly draw a self-supervised data point $x_{\sf ssl}$ from $\mathcal{D}_{\sf ssl}$ and generate a mixup feature $\lambda\, x_{\sf sl} + (1 - \lambda)\, x_{\sf ssl}$, where $\lambda$ is a hyperparameter of user's choice (we set $\lambda = 0.5$ in our experiments to balance the contribution from both datasets). We adopt the supervised label $y_{\sf sl}$ to provide supervised signal since the self-supervised part typically doesn't require labels.

**The Joint Optimization Objective.** Let $\theta$ denote the model parameters. Let $\ell_{\sf ssl}(x; \theta)$ denote the self-supervised loss, e.g., MSE reconstruction loss for masked autoencoders (He et al., 2022), and let $\ell_{\sf sl}(f(x), y; \theta)$ denote the supervised loss, e.g., cross-entropy for image classification (Krizhevsky et al., 2012). Let $\mathcal{D}_{\sf ssl}$ represent the SSL dataset, which contains examples $\{x_i\}_i$, and $\mathcal{D}_{\sf sl}$ represent the SL dataset, which also contains examples $\{(x_i, y_i)\}_i$. These datasets serve as the sources for self-supervised and supervised learning, respectively. The classical SSL+SL framework first optimizes $\min_\theta \mathbb{E}_{x, y \sim \mathcal{D}_{\sf ssl}}[\ell_{\sf ssl}(x; \theta)]$ and then optimizes $\min_\theta \mathbb{E}_{x, y \sim \mathcal{D}_{\sf sl}}[\ell_{\sf sl}(f(x), y; \theta)]$.

To integrate self-supervised and supervised learning objectives, MixTraining considers a weighted combination of these two objectives and optimizes the following goal:

$$\min_\theta \mathbb{E}_{(x,y) \sim \mathcal{D}_{\sf mix}}[\alpha\, \ell_{\sf ssl}(x; \theta) + (1 - \alpha)\, \ell_{\sf sl}(f(x), y; \theta)], \tag{3}$$

where $\mathcal{D}_{\sf mix}$ is the mixed dataset and $\alpha \in (0, 1)$ is a hyperparameter LOSS-RATIO designed to balance the focus between learning general representations (by optimizing self-supervised loss $\ell_{\sf ssl}(x; \theta)$) and achieving specific target (by optimizing supervised loss $\ell_{\sf sl}(f(x), y; \theta)$). Eq. (3) can be naturally extended to include a continuously annealed loss weight $\alpha$ that interpolates between the SSL and SL objectives. For simplicity, we retain the current form of Eq. (3); our ablation study in Section 4.2.4 shows that both linear and cosine annealing yield comparable performance.

## 4 Experiments

### 4.1 Setups

**Datasets.** We conduct experiments on standard computer vision datasets, including CIFAR-10 (Krizhevsky et al., 2009), CIFAR-100 (Krizhevsky et al., 2009), and TinyImageNet (Le & Yang, 2015).

**Models.** Our model consists of three components: a shared backbone model, a classification head for SL, and a reconstruction head for SSL. For most of our experiments, we use the standard ViT-Tiny (ViT-T) (Dosovitskiy et al., 2020; Wu et al., 2022) as the backbone and the classification head. We adopt masked autoencoding (MAE) (He et al., 2022) as the reconstruction task and use an MAE decoder of depth 2 as the reconstruction head. In Section 4.4, we provide additional experiments with ResNet-18, where the reconstruction task is standard autoencoding (AE) and the reconstruction head is a two-layer transposed convolution (Zeiler et al., 2010).

Table 1: Accuracy and latency comparison using the ViT-T model. We evaluate performance across various data limitation levels $p \in \{10\%, 25\%, 50\%, 75\%, 100\%\}$. The highest accuracy in each setting is highlighted in **bold** (if gain $\geq 0.5\%$). MixTraining achieves a Pareto improvement over the SSL+SL baseline, **achieving higher accuracy and lower latency in 15 out of 15 settings.**

| Datasets | Methods | 10% | | 25% | | 50% | | 75% | | 100% | |
|---|---|---|---|---|---|---|---|---|---|---|---|
| | | Accuracy↑ | Latency(s)↓ | Accuracy↑ | Latency(s)↓ | Accuracy↑ | Latency(s)↓ | Accuracy↑ | Latency(s)↓ | Accuracy↑ | Latency(s)↓ |
| TinyImageNet | SL | 7.24% | 1102.42 | 20.91% | 2619.14 | 30.36% | 5181.44 | 36.98% | 7709.21 | 41.96% | 10257.12 |
| | SSL+SL | 10.21% | 2382.06 | 21.78% | 5813.27 | 34.30% | 11449.43 | 42.73% | 17296.22 | 46.65% | 22917.29 |
| | MixTraining | **20.99%** | 1913.75 | **31.43%** | 4516.69 | **42.77%** | 8868.88 | **49.30%** | 13304.60 | **55.46%** | 17795.47 |
| CIFAR-10 | SL | 53.40% | 589.44 | 66.03% | 1335.79 | 75.00% | 2593.27 | 79.22% | 3844.27 | 81.52% | 5155.59 |
| | SSL+SL | 56.79% | 1253.65 | 67.95% | 2774.24 | 77.06% | 5354.08 | 82.11% | 7993.72 | 84.69% | 10730.52 |
| | MixTraining | **60.45%** | 962.16 | **72.61%** | 2206.13 | **79.95%** | 4247.78 | **83.97%** | 6234.03 | **87.13%** | 8274.45 |
| CIFAR-100 | SL | 19.05% | 609.11 | 31.49% | 1346.96 | 42.50% | 2580.08 | 48.97% | 3814.54 | 54.72% | 5022.96 |
| | SSL+SL | 22.27% | 1279.78 | 34.93% | 2882.62 | 46.02% | 5551.88 | 53.79% | 8226.83 | 57.92% | 10960.08 |
| | MixTraining | **25.19%** | 1010.94 | **38.55%** | 2226.92 | **48.60%** | 4298.75 | **55.84%** | 6354.76 | **59.95%** | 8457.93 |

**Baselines.** We evaluate the performance of our algorithm (Algorithm 1) against the following baselines:

- *Supervised learning (SL).* Conduct standard supervised learning on the backbone and the classification head with cross-entropy loss for $e_{\mathsf{sl}}$ epochs.

- *Self-supervised learning + supervised learning (SSL+SL).* Conduct self-supervised learning on the backbone and the reconstruction head with MSE loss for $e_{\mathsf{ssl}}$ epochs and then conduct standard supervised learning with cross-entropy loss for $e_{\mathsf{sl}}$ epochs.

The comparison between SL and SSL+SL reflects the compute-performance trade-off: SSL+SL achieves better performance at the cost of the added computation in the SSL-alone step. We aim to provide a better compute-performance trade-off with MixTraining, i.e., a Pareto improvement over the SSL+SL baseline. We adopt a standard set of augmentations following the practices of He et al. (2016); Cubuk et al. (2020); Charisoudis et al. (2023), as detailed in Appendix A.3. We remark that our goal is not to maximize benchmark accuracy, but to enable a controlled and reproducible comparison across different training pipelines.

**Evaluation Metrics.** For each method, we measure its performance by the accuracy on the downstream classification task and its computation cost by the training latency (i.e., the total wall-clock time). We report the average accuracy and latency over 4 runs with different random seeds. We calculate the speedups of our method as the ratio between the latency of SSL+SL and MixTraining.

**Other Implementation Details.** We conduct experiments across various data limitation levels by randomly select a fraction of $p$ data points from the original dataset; we choose $p \in \{10\%, 25\%, 50\%, 75\%, 100\%\}$. In our main experiments, we set training epoch $e_{\mathsf{sl}} = e_{\mathsf{ssl}} = 100$, LOSS-RATIO $\alpha = 0.5$, and MIX-RATIO $\rho = 0.5$; we conduct detailed parameter studies for these quantities in Appendix A.6. We defer additional experimental details to Appendix A.

## 4.2 Main Results

We present our main experiment results in this section. Results across different data limitation levels are provided in Section 4.2.1, and results with varying SSL and SL datasets are presented in Section 4.2.2.

### 4.2.1 Performance Analysis across Various Data Limitation Levels

In this section, we evaluate the performance of MixTraining across various data limitation levels. We conduct experiments on CIFAR-10, CIFAR-100, and TinyImageNet datasets and present results in Table 1. As expected, the comparison between SSL+SL and SL reflects a compute-performance trade-off: SSL+SL achieves better accuracy at the cost of higher training latency. Our MixTraining method achieves a Pareto improvement over the SSL+SL baseline: **MixTraining achieves higher accuracy and lower latency in 15 out of 15 settings.**

Table 2: Accuracy and latency comparison with different self-supervised and supervised datasets using the ViT-T model. We use TinyImageNet as the self-supervised learning dataset and use CIFAR-10/CIFAR-100 as the supervised learning dataset. The highest accuracy in each setting is highlighted in **bold** (if gain $\geq 0.5\%$). MIXTRAINING achieves a Pareto improvement over the SSL+SL baseline, **achieving higher accuracy and lower latency in both settings.**

| Methods | TinyImageNet to CIFAR-10 | | TinyImageNet to CIFAR-100 | |
|---|---|---|---|---|
| | Accuracy↑ | Latency(s)↓ | Accuracy↑ | Latency(s)↓ |
| **SL** | 81.76% | 5203.76 | 54.22% | 5149.24 |
| **SSL+SL** | 84.48% | 16815.33 | 56.22% | 16582.02 |
| **MixTraining** | **89.19%** | 12098.50 | **58.49%** | 13742.15 |

Compared to baselines, MIXTRAINING achieves significant accuracy gains: for instance, on the full TinyImageNet dataset with the ViT-T model, MIXTRAINING achieves 18.89% relative accuracy gain (8.81% absolute accuracy gain) over SSL+SL and 32.17% relative accuracy gain (13.50% absolute accuracy gain) over SL. The accuracy gains are more significant under limited data: for instance, on the TinyImageNet dataset and at data limitation level of 10%, MIXTRAINING achieves 105.58% relative accuracy gain (10.78% absolute accuracy gain) over SSL+SL and 189.92% relative accuracy gain (13.75% absolute accuracy gain) over SL. MIXTRAINING also saves more compute (reflected as training latency) compared to SSL+SL. For instance, on the full TinyImageNet dataset with the ViT-T model, MIXTRAINING acheives 1.29× speedup compared to SSL+SL. These results show that MIXTRAINING provides a better compute-performance trade-off compared to the standard SSL+SL pipeline across various data limitation levels.

### 4.2.2 Self-Supervised Learning and Supervised Learning on Different Datasets

In this section, we evaluate the performance of MIXTRAINING in settings where the self-supervised learning dataset and supervised learning dataset are different. We perform self-supervised learning on the TinyImageNet dataset and supervised learning on the CIFAR-10 or CIFAR-100 datasets. We present the results in Table 2. The comparison between SSL+SL and SL still reflects the compute-performance trade-off: SSL+SL achieves better accuracy at the cost of higher training latency. Our MIXTRAINING method achieves a Pareto improvement over the SSL+SL baseline: **MixTraining achieves higher accuracy and lower latency on both settings.**

Compared to baselines, MIXTRAINING achieves significant accuracy gains: for instance, on the CIFAR-10 dataset with the ViT-T model, MIXTRAINING achieves 5.58% relative accuracy gain (4.71% absolute accuracy gain) over SSL+SL and 9.09% relative accuracy gain (7.43% absolute accuracy gain) over SL. In terms of computation cost (reflected as training latency), MIXTRAINING also saves more compute compared to SSL+SL. For instance, on the CIFAR-10 dataset with the ViT-T model, MIXTRAINING can achieve 1.39× speedup compared to SSL+SL. These results show that MIXTRAINING can provide a better compute-performance trade-off compared to the standard SSL+SL pipeline, even when using different datasets for self-supervised learning and supervised learning.

### 4.2.3 MixTraining is More Powerful than Standard Data Mixups

To examine the advantages of MIXTRAINING over standard data mixups (Zhang et al., 2017), we further equip the SL and SSL+SL baselines with mixup data augmentation (Table 3) on both CIFAR-10 and CIFAR-100 datasets. While mixup improves the baselines' performance, our proposed MIXTRAINING method consistently achieves the best performance; see Table 15 for results across different data levels. This demonstrates that MIXTRAINING provides advantages that standard mixup alone cannot achieve—the gains stem from the unified MIXTRAINING pipeline described in Section 3.2, rather than just from augmentation effects.

Table 3: Accuracy comparison on CIFAR-10 and CIFAR-100 with mixup augmentations applied to the SL and SSL+SL baselines (with ViT-T). We report results under the 100% data setting. The highest accuracy is highlighted in **bold** (if gain $\geq 0.5\%$). MixTraining achieves the best performance over the baselines.

| Datasets | CIFAR-10 | CIFAR-100 |
|---|---|---|
| **SL w/ mixup** | 83.87% | 58.96% |
| **SSL+SL w/ mixup** | 85.81% | 59.41% |
| **MixTraining** | **87.13%** | **59.95%** |

### 4.2.4 Smooth Interpolation of $\alpha$ in the Mixtraining Phase

While Eq. (3) uses a fixed loss weight $\alpha \in [0, 1]$, it can be naturally extended to include a continuously annealed weight $\alpha$ that interpolates between the SSL and SL objectives. In Table 4, we conduct ablations with both linear and cosine annealing schedules. Since both variants achieve comparable performance to our original MixTraining approach, this indicates that the key advantage lies in the existence of the mixtraining phase itself, rather than in the specific scheduling of the weight $\alpha$.

Table 4: Ablation of linear and cosine annealing schedules for loss weight $\alpha$ in MixTraining. We conduct experiments on the CIFAR-10 dataset with the ViT-T model. All variants achieve comparable performance.

| Methods | 10% | 25% | 50% | 75% | 100% |
|---|---|---|---|---|---|
| **Algorithm 1** | 60.45% | 72.61% | 79.95% | 83.97% | 87.13% |
| **Algorithm 1 w/ linear** | 61.49% | 72.21% | 79.07% | 84.82% | 86.90% |
| **Algorithm 1 w/ cosine** | 60.65% | 72.03% | 79.16% | 83.40% | 87.00% |

### 4.2.5 Impact of Training Epochs

We study the impact of varying training epochs $e_{\mathsf{ssl}}$, $e_{\mathsf{sl}}$ on model accuracy and training latency. For simplicity, we set $e_{\mathsf{ssl}} = e_{\mathsf{sl}}$ and choose its value from $\{50, 75, 100\}$.[1] We conduct experiments across various choices of data limitation levels $p \in \{10\%, 25\%, 50\%, 75\%, 100\%\}$. Due to space limitations, we present results for learning with full data ($p = 100\%$) in the main content, and defer the complete results to Appendix A.4.

We present results with full data in Table 5. We observe that, compared to SL and SSL+SL, MixTraining generally yields greater accuracy gains for added computation; for example, doubling the training epochs from 50 to 100 results in a larger improvement in accuracy. Interestingly, MixTraining with $e_{\mathsf{ssl}} = e_{\mathsf{sl}} = 50$ outperforms SL with $e_{\mathsf{sl}} = 100$ and SSL+SL with $e_{\mathsf{ssl}} = e_{\mathsf{sl}} = 100$ in terms of both accuracy and compute: MixTraining simultaneously achieves higher accuracy and lower training latency. MixTraining consistently acheives a Pareto improvement over the SSL+SL baseline: **MixTraining achieves higher accuracy and lower latency in all 9 out of 9 settings**. Combining results in Appendix A.4 with data limitations levels $p \in \{25\%, 50\%, 75\%, 100\%\}$, **MixTraining achieves higher accuracy and lower latency in all 45 out of 45 settings.** These improvements again demonstrate that MixTraining is a superior training pipeline compared to the standard SSL+SL approach.

## 4.3 MixTraining in New Settings

### 4.3.1 Learning in Semi-Supervised Settings

We extend our experiments to the semi-supervised setting where 50% data are labeled and the other 50% data are unlabeled. In this setup, the SL baseline trains only on the labeled data, while the SSL+SL baseline

---

[1]When $e_{\mathsf{ssl}} = e_{\mathsf{sl}} = 100$, all validation loss curves have converged. Some validation losses may not fully converge when the number of training epochs is reduced; however, this is intentional, as we aim to study compute-performance trade-offs under compute-constrained settings.

Table 5: Parameter study on training epochs with full data using the ViT-T model. Results under other data limitation levels $p \in \{10\%, 25\%, 50\%, 75\%\}$ are deferred to Appendix A.4. The highest accuracy in each setting is highlighted in **bold** (if gain $\geq 0.5\%$). MixTraining achieves a Pareto improvement over the SSL+SL baseline, **achieving higher accuracy and lower latency in 9 out of 9 settings.**

| Datasets | Computation (epoch) | SL | | SSL+SL | | MixTraining | |
|---|---|---|---|---|---|---|---|
| | | Accuracy↑ | Latency(s)↓ | Accuracy↑ | Latency(s)↓ | Accuracy↑ | Latency(s)↓ |
| **TinyImageNet** | 50 | 39.75% | 4957.82 | 42.16% | 10714.36 | **50.96%** | 8410.97 |
| | 75 | 41.74% | 7560.40 | 44.22% | 16498.39 | **53.09%** | 13111.49 |
| | 100 | 41.96% | 10257.12 | 46.65% | 22917.29 | **55.46%** | 17795.47 |
| **CIFAR-10** | 50 | 80.78% | 2489.38 | 83.75% | 5365.33 | **85.60%** | 4193.42 |
| | 75 | 81.59% | 3818.88 | 84.48% | 8070.96 | **86.79%** | 6515.22 |
| | 100 | 81.52% | 5155.59 | 84.69% | 10730.52 | **87.13%** | 8274.45 |
| **CIFAR-100** | 50 | 54.25% | 2713.55 | 56.91% | 5763.71 | **58.67%** | 4421.63 |
| | 75 | 54.71% | 3881.84 | 56.95% | 8104.50 | **59.11%** | 6459.23 |
| | 100 | 54.72% | 5022.96 | 57.92% | 10960.08 | **59.95%** | 8457.93 |

Table 6: Accuracy comparison in semi-supervised setting with 50% labeled data and 50% unlabeled data. We conduct experiments on the CIFAR-10 dataset with the ViT-T model. MixTraining achieves the **best** result over the baselines.

| Metric | SL | SSL+SL | MixTraining |
|---|---|---|---|
| **Accuracy↑** | 75.59 | 83.89 | **87.35** |

first applies reconstruction-based SSL on all data and then performs supervised learning on the labeled subset. In our MixTraining framework, the SSL and SL phases remain the same as in the standard SSL+SL pipeline, while the intermediate mixtraining phase randomly mixes labeled data and train using the objective in Eq. (3). As shown in Table 6, MixTraining achieves the highest accuracy in this semi-supervised setting, with even larger gains over the baselines. This further highlights the effectiveness of MixTraining.

### 4.3.2 Learning with Data Noise

We simulate a real-world environment by adding random label noise to the dataset: for each data point, with probability $p = 0.2$, we randomly change its label. Table 7 shows that our MixTraining framework continues to achieve the best performance with data noise, verifying its robustness in practical settings.

### 4.4 When MixTraining Helps and When It Doesn't

We conduct additional experiments with ResNet-18 on CIFAR-10 (Table 8), a setting where the SL baseline is already highly optimized. Across all data limitation levels, MixTraining consistently improves upon the SSL+SL baseline. However, when SSL+SL (with an added SSL stage) underperforms a strong SL baseline, MixTraining—since it also includes SSL—does not fully recover SL performance. In contrast, in regimes where SSL+SL improves over SL (e.g., under limited data), MixTraining further improves upon SSL+SL.

These results refine—but do not contradict—our main conclusions. MixTraining consistently improves upon SSL+SL and offers a favorable compute–performance trade-off; yet in settings where SL is already highly optimized and adding SSL hurts performance, MixTraining may not surpass SL.

## 5 Conclusion and Future Work

We introduced MixTraining, an innovative framework that interleaves multiple self-supervised learning (SSL) and supervised learning (SL) epochs within a unified training phase, enabling a smooth transition between the two learning objectives. By enhancing the synergy between SSL and SL, MixTraining achieves significant accuracy improvements while consolidating shared computation steps to reduce computational cost. Extensive experiments demonstrate that MixTraining offers a superior compute-performance trade-off

Table 7: Accuracy comparison on CIFAR-10 (with ViT-T) with random data noise. We evaluate performance across various data limitation levels $p \in \{10\%, 25\%, 50\%, 75\%, 100\%\}$. The highest accuracy is highlighted in **bold** (if gain $\geq 0.5\%$). MixTraining achieves the best performance over the baselines.

| Methods | 10% | 25% | 50% | 75% | 100% |
|---|---|---|---|---|---|
| **SL** | 55.80% | 64.64% | 72.96% | 78.33% | 81.27% |
| **SSL+SL** | 56.61% | 68.45% | 72.94% | 80.98% | 84.17% |
| **MixTraining** | **61.18%** | **70.44%** | **75.97%** | **84.01%** | **86.13%** |

Table 8: Accuracy and latency comparison using the ResNet-18 model. We evaluate performance across various data limitation levels $p \in \{10\%, 25\%, 50\%, 75\%, 100\%\}$. The highest accuracy in each setting is highlighted in **bold** (if gain $\geq 0.5\%$). MixTraining consistently improves over SSL+SL but may not match the SL performance if SSL+SL degrades performance relative to a strong SL baseline.

| Datasets | Methods | 10% | 25% | 50% | 75% | 100% |
|---|---|---|---|---|---|---|
| | **SL** | 60.42 | 79.86 | 85.31 | 87.97 | **92.05** |
| **CIFAR-10** | **SSL+SL** | 65.29 | 82.20 | 85.24 | 87.12 | 87.18 |
| | **MixTraining** | **67.33** | **82.73** | 85.30 | 87.63 | 87.91 |
| | **SL** | 28.76 | 40.67 | 50.25 | 56.56 | **77.87** |
| **CIFAR-100** | **SSL+SL** | 28.57 | 40.78 | 51.44 | 55.73 | 59.76 |
| | **MixTraining** | **29.81** | **41.76** | **52.30** | **57.62** | 61.54 |

compared to the conventional SSL+SL pipeline: MixTraining achieves substantial improvements in model accuracy while significantly reducing training latency.

While our work focuses on scenarios where practitioners need to train models from scratch due to data privacy constraints or domain-specific requirements, we believe MixTraining can also be leveraged in larger-scale settings when positioned as a *mid-training* strategy, rather than as a replacement for fully task-agnostic pretraining. As an initial validation of this idea, we conduct an additional experiment where a ViT-T model pretrained on TinyImageNet is adapted to CIFAR-10. Compared to directly fine-tuning the pretrained model (84.48% accuracy), applying MixTraining on the target dataset (as pretraining data is typically unavailable in this case) improves performance to 88.75%, indicating that MixTraining can preserve general representations while enhancing task-specific adaptation. We believe this mid-training perspective opens an interesting direction for future work.

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

# A  Additional Implementation Details and Experimental Results

## A.1  Software and Hardware Dependencies

All of the codes are based on PyTorch[2] (Paszke et al., 2019) with timm library[3] (Wightman et al., 2021). All experiments in this paper are conducted on one NVIDIA RTX 6000 Ada GPU.

## A.2  Computation Reuse for Different Self-Supervised Learning Approaches

**MAE.**  To enable compute reuse in MIXTRAINING, we use a modified version of MAE. All input patches are first processed by the backbone to produce feature representations. For the classification head, we use all feature representations directly. For the reconstruction head, we randomly mask a subset of the feature representations according to a predefined mask ratio and compute the reconstruction loss only on the masked patches. This modification changes the reconstruction behavior of MAE but leaves the supervised learning stage unchanged, which is the basis for performance evaluation in our experiments.

**Standard AE.**  Computation reuse in the standard AE approach is straightforward: the backbone outputs are fed directly to both the reconstruction head and the classification head.

## A.3  Implementation Details of Computer Vision Models

**ViT-T.**  Our ViT-T implementations are largely derived from the seminal work from Wu et al. (2022) and shrink the decoder to 2 layers as suggested in the reproduction challenge by Charisoudis et al. (2023). Specifically, we set the embedding dimension ('emb_dim') to 192, with the encoder and decoder configured to 12 and 2 layers respectively, alongside 3 heads each for both encoder and decoder. The masking ratio is maintained at 0.75 as He et al. (2022) suggest. For the CIFAR-10, CIFAR-100 datasets, the image resolution is standardized to $32 \times 32$ pixels with a patch size of 2, while the image resolution is $64 \times 64$ pixels with a patch size of 4 for TinyImageNet to ensure uniform computational complexity across all experiments.

**ResNet-18.**  Our ResNet-18 implementations are derived from the seminal work from He et al. (2016) on deep residual networks. To facilitate self-supervised learning through autoencoders (AE), we augment the ResNet architectures with two additional two-dimensional transposed convolution layers post-backbone for the reconstruction task.

**Data Augmentation and Preprocessing.**  For the preprocessing of CIFAR-10 and CIFAR-100, we adopt simple preprocessing as in He et al. (2016), which randomly crops the images to a size of $32 \times 32$ pixels, with a padding of 4 pixels on each side of the image, then randomly flips the images horizontally with a 50% probability. For TinyImageNet, we apply random horizontal flip (probability 0.5), random resized crop (scale $(0.6, 1.0)$ and ratio $(0.9, 1.1)$), and RandAug(2,9) (Cubuk et al., 2020; Charisoudis et al., 2023) for data augmentation.

**Hyperparameters.**  Detailed training hyperparameters used in our experiments are summarized in Table 9 (ViT-T) and Table 10 (ResNet-18). For experiments on the TinyImageNet dataset using ViT-T, following Charisoudis et al. (2023), we slightly modify the hyperparameters in Table 9: we set the base learning rate to $1 \times 10^{-3}$ and $2 \times 10^{-3}$, the Adam betas to $(0.9, 0.95)$ and $(0.9, 0.999)$, and the weight decay to 0.15 and 0.05 for self-supervised and supervised learning, respectively.

## A.4  Additional Experiment Results

We report additional experimental results across various data limitation levels and training epochs. Results with full data are provided in main text (Appendix A.6), and results with $p \in \{10\%, 25\%, 50\%, 75\%\}$ data limitation levels are presented in Tables 11 to 14. These results show that MIXTRAINING provides a better

---

[2]https://pytorch.org/
[3]https://huggingface.co/timm

Table 9: Hyperparameters for ViT-T across self-supervised and supervised settings.

| Hyperparameters | Self-Supervised learning | Supervised Learning |
|---|---|---|
| Batch Size | 256 | 256 |
| Base Learning Rate | $1.5 \times 10^{-4}$ | $1 \times 10^{-3}$ |
| Learning Rate Scheduler | CosineAnnealing | CosineAnnealing |
| Optimizer | AdamW | AdamW |
| Betas | (0.9, 0.95) | (0.9, 0.95) |
| Weight Decay | 0.05 | 0.05 |
| Warmup Epoch | 20 | 5 |

Table 10: Hyperparameters for ResNet-18 across self-supervised and supervised settings.

| Hyperparameters | Self-Supervised learning | Supervised Learning |
|---|---|---|
| Batch Size | 256 | 256 |
| Base Learning Rate | $1 \times 10^{-4}$ | $1 \times 10^{-3}$ |
| Learning Rate Scheduler | CosineAnnealing | CosineAnnealing |
| Optimizer | AdamW | AdamW |
| Betas | (0.9, 0.95) | (0.9, 0.95) |
| Weight Decay | 0.05 | 0.05 |
| Warmup Epoch | 20 | 5 |

compute-performance trade-off compared to the standard SSL+SL pipeline: **MixTraining achieves higher accuracy and lower latency in all 36 out of 36 settings over SSL+SL.**

Table 11: Parameter study on training epochs under 10% data limitation using the ViT-T model. The highest accuracy in each setting is highlighted in **bold** (if gain $\geq 0.5\%$). MixTraining achieves a Pareto improvement over the SSL+SL baseline, **achieving higher accuracy and lower latency in 9 out of 9 settings.**

| Datasets | Computation (epoch) | SL | | SSL+SL | | MixTraining | |
|---|---|---|---|---|---|---|---|
| | | Accuracy↑ | Latency(s)↓ | Accuracy↑ | Latency(s)↓ | Accuracy↑ | Latency(s)↓ |
| **TinyImageNet** | 50 | 7.69% | 536.82 | 8.67% | 1136.96 | **16.53%** | 896.16 |
| | 75 | 7.62% | 809.11 | 9.89% | 1762.56 | **20.23%** | 1376.22 |
| | 100 | 7.24% | 1102.42 | 10.21% | 2382.06 | **20.99%** | 1913.75 |
| **CIFAR-10** | 50 | 53.28% | 287.83 | 54.75% | 637.27 | **57.60%** | 473.80 |
| | 75 | 53.07% | 434.86 | 55.59% | 937.01 | **58.59%** | 742.53 |
| | 100 | 53.40% | 589.44 | 56.79% | 1253.65 | **60.45%** | 962.16 |
| **CIFAR-100** | 50 | 19.08% | 294.14 | 20.26% | 677.26 | **23.36%** | 516.54 |
| | 75 | 19.39% | 464.97 | 21.15% | 981.43 | **24.27%** | 770.75 |
| | 100 | 19.05% | 609.11 | 22.27% | 1279.78 | **25.19%** | 1010.94 |

## A.5 Comparison of Baselines with Mixup Augmentations across Different Data Levels

We present in Table 15 of experiment comaprison against baselines with mixup data augmentations across different data limitation levels.

## A.6 Parameter Study

In this section, we explore the impacts of varying LOSS-RATIO $\alpha$, MIX-RATIO $\rho$, and training epochs $e_{\mathsf{ssl}}$ and $e_{\mathsf{sl}}$ for MixTraining.

Table 12: Parameter study on training epochs under 25% data limitation using the ViT-T model. The highest accuracy in each setting is highlighted in **bold** (if gain ≥ 0.5%). MIXTRAINING achieves a Pareto improvement over the SSL+SL baseline, **achieving higher accuracy and lower latency in 9 out of 9 settings.**

| Datasets | Computation (epoch) | SL | | SSL+SL | | MixTraining | |
|---|---|---|---|---|---|---|---|
| | | Accuracy↑ | Latency(s)↓ | Accuracy↑ | Latency(s)↓ | Accuracy↑ | Latency(s)↓ |
| **TinyImageNet** | 50 | 17.60% | 1268.41 | 17.31% | 2771.72 | **31.69%** | 2129.89 |
| | 75 | 19.49% | 1921.23 | 19.80% | 4223.24 | **31.90%** | 3281.00 |
| | 100 | 20.91% | 2619.14 | 21.78% | 5813.27 | **31.43%** | 4516.69 |
| **CIFAR-10** | 50 | 65.03% | 646.45 | 67.65% | 1409.32 | **70.69%** | 1076.16 |
| | 75 | 65.69% | 985.54 | 68.57% | 2135.05 | **71.29%** | 1719.24 |
| | 100 | 66.03% | 1335.79 | 67.95% | 2774.24 | **72.61%** | 2206.13 |
| **CIFAR-100** | 50 | 30.87% | 651.25 | 34.46% | 1517.26 | **36.87%** | 1161.89 |
| | 75 | 31.21% | 1033.91 | 35.05% | 2237.72 | **38.43%** | 1720.56 |
| | 100 | 31.49% | 1346.96 | 34.93% | 2882.62 | **38.55%** | 2226.92 |

Table 13: Parameter study on training epochs under 50% data limitation using the ViT-T model. The highest accuracy in each setting is highlighted in **bold** (if gain ≥ 0.5%). MIXTRAINING achieves a Pareto improvement over the SSL+SL baseline, **achieving higher accuracy and lower latency in 9 out of 9 settings.**

| Datasets | Computation (epoch) | SL | | SSL+SL | | MixTraining | |
|---|---|---|---|---|---|---|---|
| | | Accuracy↑ | Latency(s)↓ | Accuracy↑ | Latency(s)↓ | Accuracy↑ | Latency(s)↓ |
| **TinyImageNet** | 50 | 24.95% | 2496.29 | 27.27% | 5401.40 | **40.37%** | 4280.53 |
| | 75 | 28.81% | 3827.39 | 31.15% | 8363.36 | **41.99%** | 6530.16 |
| | 100 | 30.36% | 5181.44 | 34.30% | 11449.43 | **42.77%** | 8868.88 |
| **CIFAR-10** | 50 | 73.70% | 1252.78 | 75.26% | 2703.05 | **78.83%** | 2133.61 |
| | 75 | 74.63% | 1931.89 | 76.40% | 4155.73 | **79.31%** | 3255.72 |
| | 100 | 75.00% | 2593.27 | 77.06% | 5354.08 | **79.95%** | 4247.78 |
| **CIFAR-100** | 50 | 42.32% | 1248.02 | 44.95% | 2924.08 | **48.53%** | 2246.85 |
| | 75 | 42.46% | 1981.46 | 45.23% | 4355.99 | **48.87%** | 3275.14 |
| | 100 | 42.50% | 2580.08 | 46.02% | 5551.88 | **48.60%** | 4298.75 |

Table 14: Parameter study on training epochs under 75% data limitation using the ViT-T model. The highest accuracy in each setting is highlighted in **bold** (if gain ≥ 0.5%). MIXTRAINING achieves a Pareto improvement over the SSL+SL baseline, **achieving higher accuracy and lower latency in 9 out of 9 settings.**

| Datasets | Computation (epoch) | SL | | SSL+SL | | MixTraining | |
|---|---|---|---|---|---|---|---|
| | | Accuracy↑ | Latency(s)↓ | Accuracy↑ | Latency(s)↓ | Accuracy↑ | Latency(s)↓ |
| **TinyImageNet** | 50 | 33.16% | 3713.97 | 34.81% | 8062.47 | **46.44%** | 6369.23 |
| | 75 | 36.64% | 5708.06 | 40.31% | 12538.24 | **49.29%** | 9779.95 |
| | 100 | 36.98% | 7709.21 | 42.73% | 17296.22 | **49.30%** | 13304.60 |
| **CIFAR-10** | 50 | 78.08% | 1868.81 | 79.60% | 4033.65 | **82.91%** | 3163.51 |
| | 75 | 78.74% | 2856.25 | 81.24% | 6216.65 | **83.68%** | 4762.06 |
| | 100 | 79.22% | 3844.27 | 82.11% | 7993.72 | **83.97%** | 6234.03 |
| **CIFAR-100** | 50 | 48.99% | 2009.17 | 51.85% | 4345.47 | **54.30%** | 3329.29 |
| | 75 | 49.00% | 2929.78 | 53.25% | 6312.75 | **55.05%** | 4857.74 |
| | 100 | 48.97% | 3814.54 | 53.79% | 8226.83 | **55.84%** | 6354.76 |

Table 15: Accuracy comparison on CIFAR-10 (with ViT-T) with mixup augmentations applied to the SL and SSL+SL baselines. We evaluate performance across various data limitation levels $p \in \{10\%, 25\%, 50\%, 75\%, 100\%\}$. We conduct mixup for both of the other two baselines. The highest accuracy is highlighted in **bold** (if gain $\geq 0.5\%$). MixTraining achieves the best performance over the baselines.

| Methods | 10% | 25% | 50% | 75% | 100% |
|---|---|---|---|---|---|
| **SL w/ mixup** | 56.45% | 68.41% | 76.18% | 80.91% | 83.87% |
| **SSL+SL w/ mixup** | 57.92% | 70.08% | 77.50% | 82.83% | 85.81% |
| **MixTraining** | **60.45%** | **72.61%** | **79.95%** | **83.97%** | **87.13%** |

Table 16: Parameter study on LOSS-RATIO $\alpha$. We train the ViT-T model with full data and set $\rho = 0.75$; other experimental settings remain the same as Table 1. The best and second-best accuracies are highlighted in **bold** and underline, respectively.

| $\alpha$ | TinyImageNet | CIFAR-10 | CIFAR-100 |
|---|---|---|---|
| 0.01 | 41.58% | 79.61% | 52.58% |
| 0.1 | 53.17% | 82.19% | 55.21% |
| 0.5 | **53.86%** | 85.75% | **58.88%** |
| 0.9 | 48.18% | **86.13%** | 58.47% |
| 0.99 | 42.67% | 83.75% | 56.04% |

Table 17: Parameter study on MIX-RATIO $\rho$. We train the ViT-T model with full data and set $\alpha = 0.5$; other experimental settings remain the same as Table 1. The best and second-best accuracies are highlighted in **bold** and underline, respectively.

| $\rho$ | TinyImageNet | CIFAR-10 | CIFAR-100 |
|---|---|---|---|
| 0.25 | 55.08% | 85.49% | **60.98%** |
| 0.50 | **55.40%** | 85.54% | 60.15% |
| 0.75 | 53.86% | **85.75%** | 58.88% |
| 1.00 | 49.21% | 84.33% | 55.80% |

### A.6.1 Impact of loss-ratio $\alpha$

We study the impact of varying hyperparameter LOSS-RATIO $\alpha$ on model accuracy in this section. We conduct experiments with $\alpha \in \{0.01, 0.1, 0.5, 0.9, 0.99\}$ and report the accuracy in Table 16; we didn't report the latency since varying $\alpha$ doesn't change the overall computation cost. As shown in Table 16, LOSS-RATIO $\alpha = 0.5$ generally leads to good accuracy gains—either achieving the highest accuracy (2 out of 3) or achieving the second-best accuracy (1 out of 3). This indicates that a well-chosen $\alpha$ should appropriately balance self-supervised learning and supervised learning objectives in MixTraining. An $\alpha = 0.5$ allows the model to focus on both self-supervised learning and supervised learning objectives. Therefore, we recommend setting $\alpha = 0.5$ in experiments.

### A.6.2 Impact of mix-ratio $\rho$

We study the impact of varying the hyperparameter MIX-RATIO $\rho$ on model accuracy in this section. We conduct experiments with $\rho \in \{0.25, 0.5, 0.75, 1\}$ and report the accuracy in Table 17; we didn't report latency since it is straightforward to see that large $\rho$ reduces latency (Section 3.2). Values of $\rho = 0.5$ or $\rho = 0.25$ generally leads to good accuracy gains. Since larger $\rho$ reduces latency, its selection should be guided by individual priorities, such as accelerating the learning process or achieving higher accuracy.

