# OpenReview forum: "MixTraining: A Better Trade-Off Between Compute and Performance"
_TMLR — Accepted by TMLR_

### Review · Reviewer_Coby · 2025-10-07

**Summary Of Contributions:**

This paper introduces "MixTraining," a novel strategy for the two-phase training pipeline of self-supervised learning (SSL) followed by supervised learning (SL), or fine-tuning. The method is presented in the context of reconstruction-based SSL objectives. MixTraining introduces an intermediate phase, where the SSL and SL objectives are optimized *jointly* as a weighted sum. The authors provide an empirical validation of their approach, demonstrating its potential for improved accuracy and reduced training time on downstream tasks.

**Strength:**
The primary strength of this paper is its simple yet effective proposal of a weighted, joint optimization phase. The reported improvements in accuracy are noteworthy and suggest a promising potential for better integration of reconstruction based self-supervised and supervised learning workflow.

**Weakness:**
I believe the paper requires further strengthening by addressing a few points, primarily concerning the alignment between the stated motivation and the experimental validation. Critically, this appears to lead to partially missing empirical support for the claims made. Yet, I believe it also raises opportunities to enhance quality and clarity of the manuscript. I will outline these points in the following:

The authors formulate the **Scope and Applicability** of their work as: *“It is important to clarify that MixTraining does not aim to replace the existing pretraining-finetuning paradigm when high-quality pretrained models are readily available. Instead, our framework targets scenarios where practitioners must train models from scratch due to data privacy constraints, domain-specific requirements where generic pretrained models prove inadequate, or the absence of suitable pretrained models. In these settings, MixTraining provides a more efficient alternative to the standard SSL+SL pipeline, achieving superior performance while reducing training latency.”*  — While I find this to be a relevant motivation and problem formulation, the respective empiric investigation and results are targeting standard benchmarks such as TinyImageNet. It would be interesting to see the method evaluated in a scenario that mirrors stated scope more closely. I fear the authors choice of validation weakens some of the central claims of this work.

**(1)** Regarding the **"accuracy gains,"** the use of mixup data-augmentation in MixTraining is a neat mechanism. For the sake of a clear comparison, it would be helpful to clarify whether mixup was also applied to the SL and SSL+SL baselines?

Also, I would appreciate a bit more or deeper intuition regarding the mechanism behind the accuracy gains. The authors state that: *“Since the self-supervised learning phase aims at learning general representations and the following supervised learning phase aims at learning task-specific information, intuitively, these two phases optimize the model in different directions.”*  and “[…] *creating a smooth transition that better preserves learned representations while adapting to downstream tasks*”. -- I believe the term "different direction" needs to be defined more clearly here. Assuming it refers to the gradient direction, the positive cosine similarity shown in Figure 3 is an interesting intuition. Yet, it might be beneficial to discuss whether a negative cosine similarity between the “different directions” is possible, and if this intuition holds true across different layers of the model, e.g. layers at different depth. My initial thought was that features learned for a reconstruction loss would be quite distinct from those learned for a discriminative task. Can those gradients be aligned? Or is it the diversity of the joint training objective that yields benefit here? To me, it is not obvious that features to reconstruct data are aligned to those that allow to discriminate data. After all this is why often a subsequent finetuning stage is necessary for reconstruction-based SSL methods.

**(2)** Concerning the **"computational gains,"** the authors note that: *“the merge of forward/backward passes over the backbone model allows us to reduce computation compared to the synchronous setting substantially.”* It would be helpful to clarify if this implies that MixTraining requires fewer gradient steps to converge? I believe readers would appreciate a respective ablation study. In relation to the improved “GPU utilization” shown in Figure 3, I wonder if this metric might be influenced by the smaller scale of the datasets used. Using lower resolution data could lead to utilization below 100% on modern hardware. As I understand the potential for computational efficiency gain is partly tied to under-exploited parallelism possibility on the hardware, it would be valuable to understand if the potential for increased parallelism does translate to practical benefits in training time in scenarios where GPU utilization is already saturated.

**(3)** Regarding **"the mixed dataset,"** a mixup-like mechanism is proposed for combining samples for the supervised and self-supervised objectives. Since the current dataset has labels for all samples, the unique aspects of this mixing strategy are actually not explored. An experiment where a portion of the data is unlabeled would be the more natural approach to demonstrate the effectiveness of this mechanism.

To further elaborate on the connection between motivation and implementation, I have a couple of additional thoughts:

- A key insight of the paper is that "the rigid separation between SSL and SL phases in conventional pipelines is neither necessary nor optimal." However, the proposed MixTraining is still a rigid distinct phase. It would be interesting to investigate on a more seamless transition. For example, the proposed framework readily allows for a continuously annealing of the $\alpha$ weight between the SSL and SL losses. The authors suggest that a smooth transition avoids instability, hence a more gradual approach should offer even greater benefits in this regard.
- The main direction for future work is stated as: *“An important next step is to evaluate MixTraining in larger-scale settings, using more powerful models such as ViT-L/16 (Dosovitskiy et al., 2020; Zhai et al., 2022) and larger datasets such as ImageNet-21K (Ridnik et al., 2021).”*  — While scaling data and model is typically considered a valuable research direction, it seems to be in particularly stark contrast with the initial motivation.

Finally, a few minor suggestions for clarity:

- The paper's focus on reconstruction-based SSL, such as masked autoencoders, could be stated more explicitly early on. While MixTraining is presented in the context of general self-supervised learning, the experiments are specific to reconstruction tasks. Clarifying this focus would help manage reader expectations. If the authors envision MixTraining being applicable to other SSL paradigms like contrastive learning, including some initial experiments or a discussion on this would be a valuable addition.
- When introducing the procedure for mixing SSL and SL datasets, the phrase: *“In the simple case where self-supervised learning and supervised learning use the same feature representation (i.e., xi ),[…]”*  was a little confusing. I believe "feature representation" here refers to the input modality, and rephrasing might prevent any ambiguity.

**Audience:**

Yes

**Audience Explanation:**

Improving accuracy and computational requirements by a simple and broadly applicable mechanism in the realm of (reconstruction-based) self-supervised learning will be interesting for a wide audience.

**Broader Impact Concerns:**

None.

**Claims And Evidence:**

No

**Claims Explanation:**

While the paper presents promising initial results, the claims are not consistently supported by clear and convincing evidence, primarily due to a significant mismatch between the stated motivation and the experimental validation.

**Claim: Improved Accuracy.** The paper reports accuracy gains, but the evidence is not entirely convincing. It is unclear if the use of mixup data-augmentation, was also applied to the baseline models for a fair comparison. The intuition for the accuracy gains (optimizing in "different directions" to create a smoother transition) is not clearly defined or substantiated. The provided data (Figure 3) showing positive cosine similarity between gradients seems to contradict the initial intuition that reconstruction and discrimination tasks are "distinct," which weakens the explanation for why the method works.

**Claim: Computational Gains.** The evidence for improved efficiency is underdeveloped. The paper states that merging passes reduces computation, but it's not clarified if this translates to fewer necessary gradient steps for convergence. Likewise, the evidence for improved GPU utilization is presented on small-scale datasets where utilization might naturally be low. This makes it difficult to assess if these gains would translate to practical, scenarios where GPU resources are already saturated.

**Claim: A novel mechanism for mixed (labeled/unlabeled) datasets.** This claim lacks any direct empirical support. The mixup-like mechanism is a core part of the proposed method, but it is evaluated on a fully labeled dataset (TinyImageNet), meaning the unique aspects and benefits of this approach in a semi-supervised context are not demonstrated at all.

**Requested Changes:**

I believe this paper has a promising core idea, but significant revisions are needed to strengthen claims and improve clarity. The requested changes are outlined below, from major to minor.

**Major Revisions:**

1. **Alignment of Experimental Validation with Motivation:** This is the most critical point. The paper is motivated by scenarios where large-scale pre-trained models are unavailable (due to privacy, domain-specificity, etc.), yet evaluation is conducted on a standard benchmarks. To be clear, my concern lies not with using a benchmark that is often used, but with this benchmark(s) being not fully aligned with the objectives of the presented work. On the one hand this choice partially leaves the power of the SSL objective under-explored, but it also weakens the impact of the results. I recommend experiments that more closely reflect the stated motivation. An ideal experiment would involve a semi-supervised setup where a large portion of the dataset is unlabeled. Such a setting then also demonstrates the utility of the proposed mixed-data mechanism.
2. **Clarify Fairness of Comparisons:** It needs to be stated explicitly whether the mixup data augmentation was also applied to the SL and SSL+SL baselines. If not, the comparison should be updated to ensure all methods are compared under the same data augmentation scheme.

**Minor Revisions:**
1. **Bolster Claims of Computational Gains:** The "reduction in computation" should be further substantiated, e.g. by providing a direct comparison in an ablation study. Does it translate to fewer gradient steps for convergence? And, does the increased potential for parallelism (GPU utilization) translate to practical scenarios, e.g. larger resolution data?
2. **Further discussion of the "Rigid Separation" of training phases:** The paper claims the "rigid separation" of phases is suboptimal, yet MixTraining appears as a distinct, rigid phase. I would encourage the authors to discuss this choice and perhaps explore a more seamless transition, for example, by continuously annealing the loss weight α over time.
3. **Overall Clarity and Scoping:**
    - **Scope:** To manage the reader’s expectations, I recommend to state explicitly in the introduction that the paper's experiments are focused on *reconstruction-based* SSL methods. If the method is intended to be more general, a discussion or preliminary experiment on other SSL families (e.g., contrastive, redundancy-reduction, etc.) would be a valuable addition.
    - **Future Work:** I would urge to reconsider the future work paragraph. The suggestion to scale up data and model seems to contradict the primary motivation of the paper.
    - **Typo?:** Please rephrase the sentence *“In the simple case where self-supervised learning and supervised learning use the same feature representation (i.e., xi )”* to avoid confusion, as "feature representation" appears to refer to the input modality.

---

> ### Author Response · Authors · 2025-11-11
>
> Dear Reviewer Coby,
>
> We greatly appreciate your thoughtful feedback and constructive suggestions. Below, we provide detailed responses and clarifications. We have uploaded a revised manuscript with all changes highlighted in red. Please let us know if further clarification is needed.
>
> **Responses to Major Revisions:**
>
> 1. > Alignment of Experimental Validation with Motivation: This is the most critical point. The paper is motivated by scenarios where large-scale pre-trained models are unavailable (due to privacy, domain-specificity, etc.), yet evaluation is conducted on a standard benchmarks. To be clear, my concern lies not with using a benchmark that is often used, but with this benchmark(s) being not fully aligned with the objectives of the presented work. On the one hand this choice partially leaves the power of the SSL objective under-explored, but it also weakens the impact of the results. I recommend experiments that more closely reflect the stated motivation. An ideal experiment would involve a semi-supervised setup where a large portion of the dataset is unlabeled. Such a setting then also demonstrates the utility of the proposed mixed-data mechanism.
>
> **Response:** Thank you for the valuable suggestion. We conducted additional experiments in a semi-supervised setting with 50% labeled and 50% unlabeled data on CIFAR-10 with the ViT-T model. For the SL baseline, we train using only the labeled data; for the SSL+SL baseline, we perform reconstruction-based SSL on all data and supervised learning on the labeled subset. In our MixTraining framework, the SSL and SL phases remain the same as in the standard SSL+SL pipeline, while the mixtraining phase randomly mixes labeled data following the same objective described in the paper. As shown in the table below, **MixTraining continues to achieve the highest accuracy in this semi-supervised setting, with even larger gains over the baselines**. This further highlights the effectiveness of MixTraining. We have added a discussion of this result in the revised manuscript.
>
> **Semi-Supervised Results (50% labeled and 50% unlabeled)**
> | **Metric** | **SL** | **SSL+SL** | **MixTraining** |
> |-------------|:------:|:------:|:---------------:|
> | **Accuracy↑** | 75.59% | 83.89% | **87.35%** |
>
> 2. > Clarify Fairness of Comparisons: It needs to be stated explicitly whether the mixup data augmentation was also applied to the SL and SSL+SL baselines. If not, the comparison should be updated to ensure all methods are compared under the same data augmentation scheme.
>
> **Response:** Thank you for the valuable feedback. In our experiments, we adopt a simple set of augmentations following the practices of He et al. (2016) and Charisoudis et al. (2023), which doesn’t include mixup data augmentation.
>
> To address your concern, we conducted additional experiments by applying mixup augmentation to the SL and SSL+SL baselines on CIFAR-10 (with ViT-T). While mixup slightly improves the baseline results, our proposed MixTraining method consistently achieves the best performance across all data levels. This demonstrates that the observed improvements are not due to specific augmentations, but stem from the inherent advantages of our unified MixTraining pipeline, thus reinforcing our main conclusions. We have added a discussion of this result in the revised manuscript.
>
> | **Baselines** | **10% Accuracy↑** | **25% Accuracy↑** | **50% Accuracy↑** | **75% Accuracy↑** | **100% Accuracy↑** |
> |----------------|:----------------:|:-----------------:|:-----------------:|:-----------------:|:------------------:|
> | **SL w/ mixup** | 56.45% | 68.41% | 76.18% | 80.91% | 83.87% |
> | **SSL+SL w/ mixup** | 57.92% | 70.08% | 77.50% | 82.83% | 85.81% |
> | **MixTraining** | **60.45%** | **72.61%** | **79.95%** | **83.97%** | **87.13%** |

---

> ### Author Response · Authors · 2025-11-11
> **Rebuttal Continued**
>
> **Responses to Minor Revisions:**
>
> 1. > Bolster Claims of Computational Gains: The "reduction in computation" should be further substantiated, e.g. by providing a direct comparison in an ablation study. Does it translate to fewer gradient steps for convergence? And, does the increased potential for parallelism (GPU utilization) translate to practical scenarios, e.g. larger resolution data?
>
> **Response:** Thank you for the insightful comments. Our results show that, compared to the standard SSL+SL framework, the proposed MixTraining achieves higher accuracy with fewer gradient steps, thanks to the new mixtraining phase. We report latency (wall-clock training time) in our experiments (e.g., see Tables 1-4 and others in the Appendix), which directly captures the computational efficiency gains.
>
> While part of the efficiency improvement arises from better parallelism, we note that only the forward and backward passes of the SL and SSL heads benefit from parallel execution, but not the backbone computation (as illustrated in Fig. 3). The backbone-side efficiency gain instead comes from mixing data and objectives, which effectively reduces redundant backbone passes while keeping memory requirements comparable to the standard SSL+SL pipeline. Since backbone computation typically dominates overall cost, these gains translate well to practical scenarios, such as training with higher-resolution inputs.
>
> To provide a more direct comparison, we analyze the FLOPs of the two frameworks. Assuming $e_{ssl} = e_{sl}$ and that most computation lies in the shared backbone (which dominates over the SL/SSL heads), and letting $\rho \in (0,1)$ denote the mixup ratio, the total FLOPs of MixTraining are approximately $(2 - \rho)/2$ of those in the standard SL+SSL pipeline. For instance, when $\rho = 0.5$, MixTraining requires only 75% of the FLOPs compared to SSL+SL.
>
> 2. > Further discussion of the "Rigid Separation" of training phases: The paper claims the "rigid separation" of phases is suboptimal, yet MixTraining appears as a distinct, rigid phase. I would encourage the authors to discuss this choice and perhaps explore a more seamless transition, for example, by continuously annealing the loss weight α over time.
>
> **Response:** Thank you for the insightful comments. In fact, our MixTraining framework can be naturally extended to incorporate a continuously annealed loss weight $\alpha$. We conducted additional experiments on CIFAR-10 (with ViT-T) using both linear and cosine annealing schedules. As shown in the table below, both variants achieve comparable performance to our original MixTraining approach. This indicates that the key advantage lies in the existence of the mixtraining phase itself, rather than in the specific scheduling of $\alpha$. We have added a discussion of this result in the revised manuscript.
>
> | **Method** | **10% Accuracy↑** | **25% Accuracy↑** | **50% Accuracy↑** | **75% Accuracy↑** | **100% Accuracy↑** |
> |-------------|:----------------:|:-----------------:|:-----------------:|:-----------------:|:------------------:|
> | **MixTraining (Original)** | 60.45% | 72.61% | 79.95% | 83.97% | 87.13% |
> | **MixTraining (Linear)**   | 61.49% | 72.21% | 79.07% | 84.82% | 86.90% |
> | **MixTraining (Cosine)**   | 60.65% | 72.03% | 79.16% | 83.40% | 87.00% |
>
> 3. > Overall Clarity and Scoping:
> Scope: To manage the reader’s expectations, I recommend to state explicitly in the introduction that the paper's experiments are focused on reconstruction-based SSL methods. If the method is intended to be more general, a discussion or preliminary experiment on other SSL families (e.g., contrastive, redundancy-reduction, etc.) would be a valuable addition.
> Future Work: I would urge to reconsider the future work paragraph. The suggestion to scale up data and model seems to contradict the primary motivation of the paper.
> Typo?: Please rephrase the sentence “In the simple case where self-supervised learning and supervised learning use the same feature representation (i.e., xi )” to avoid confusion, as "feature representation" appears to refer to the input modality.
>
> **Response:** Thank you very much for your thoughtful comments. We have explicitly mentioned reconstruction-based SSL in the introduction, modified the future work paragraph and corrected the typos in the updated manuscript.

---

> ### Author Response · Authors · 2025-11-11
> **Rebuttal Continued**
>
> **Responses to Additional Questions:**
>
> 1. > The authors state that: “Since the self-supervised learning phase aims at learning general representations and the following supervised learning phase aims at learning task-specific information, intuitively, these two phases optimize the model in different directions.”  and “[…] creating a smooth transition that better preserves learned representations while adapting to downstream tasks”. -- I believe the term "different direction" needs to be defined more clearly here. Assuming it refers to the gradient direction, the positive cosine similarity shown in Figure 3 is an interesting intuition. Yet, it might be beneficial to discuss whether a negative cosine similarity between the “different directions” is possible, and if this intuition holds true across different layers of the model, e.g. layers at different depth. My initial thought was that features learned for a reconstruction loss would be quite distinct from those learned for a discriminative task. Can those gradients be aligned? Or is it the diversity of the joint training objective that yields benefit here? To me, it is not obvious that features to reconstruct data are aligned to those that allow to discriminate data. After all this is why often a subsequent finetuning stage is necessary for reconstruction-based SSL methods.
>
> **Response:** Thank you for your insightful comments. To further investigate this point, we conducted additional experiments on CIFAR-10 (with ViT-T) to measure the cosine similarity of gradients from different learning objectives. Our results show that the average cosine similarity between SSL and SL objectives is 0.018, indicating that these two objectives are nearly orthogonal. In contrast, the cosine similarity between SSL and MixTraining objectives is 0.791, and between SL and MixTraining objectives is 0.598, suggesting that the MixTraining objective indeed provides a smooth transition from SSL to SL. These findings support the intuition illustrated in the right plot of Fig. 2. We have added a discussion of this result in the revised manuscript.

---

> ### Comment · Reviewer_Coby · 2026-01-25
>
> I would like to thank the authors for the updated manuscript. I appreciate the inclusion of several new experiments on the CIFAR-10 benchmark, which help to corroborate the paper’s original claims and address some of the doubts expressed during my initial review:
>
> - There are additional comparisons of "MixTraining" against the "SL" and "SSL+SL" baselines including the use of mixup data-augmentation.
> - Additional experiments showcasing the use of a dynamic loss-weight $\alpha$ in comparison to the initial, more static, 3-training-phase setup, showcasing that for reasonable values of alpha the addition of the MixTraining phase appears to be more integral then the actual selection of the hyper-parameter (whether static or continuous).
> - The inclusion of an explicit experimental setup with partially labeled data on CIFAR-10 effectively demonstrates that MixTraining can achieve improved accuracy in scenarios where not all data is labeled, strengthening the paper's original motivation.
> - The authors have also provide additional intuition for the observed gradient alignment during the MixTraining phase.
>
>
> That being said, a few points remain that I would appreciate the authors' comments on:
>
> 1.  I recognize and appreciate the more nuanced pointers towards "reconstruction-based SSL" in the revised text. However, to further enhance clarity and manage reader expectations, I would still recommend explicitly stating in the introduction that the paper's scope is focused on reconstruction-based SSL and does not cover other popular SSL paradigms.
>
> 2. The revised manuscript presents several important findings in Section “4.4 Additional Results". For instance, the comparison involving mixup augmentation and the investigation into the model's robustness to the loss-weighting hyper-parameter α are crucial for validating the method. Placing these results in a separate section (and “only” for CIFAR-10) may understate their importance. Also, regarding the loss weighting, the robustness to $\alpha$ (static or continuous) raises the question whether the cosine schedule for $\alpha$ would be a preferable choice as it would remove the need for hyper-parameter $\alpha$ altogether?
>
> 3. I came to share the concern raised by Reviewer **Dw8m** regarding the relatively low accuracies obtained by the "SL" and "SSL+SL" baselines. The authors added a footnote stating that they confirmed the convergence of the training loss. While helpful, this might point to other factors limiting generalization on the test set, and the convergence statement could be strengthened by providing the loss and validation curve in the appendix. I am unsure about the the ViT-tiny architecture as this architecture type might require a lot of data and training to overcome the missing inductive bias of convolutional based architectures. For the ResNet18, I would be interested to know whether the first convolutional layer was adjusted to account for the lower resolution of CIFAR-like datasets (switching from 7x7 to a 3x3 kernel and removing the strides)?

---

> ### Author Response · Authors · 2026-02-11
>
> Dear Reviewer Coby,
>
> We greatly appreciate your follow-up comments. Below, we provide detailed responses and clarifications. A revised manuscript has also been uploaded. Please let us know if further clarification is needed.
>
>
> **Response to follow-up comments:**
>
> 1. > I recognize and appreciate the more nuanced pointers towards "reconstruction-based SSL" in the revised text. However, to further enhance clarity and manage reader expectations, I would still recommend explicitly stating in the introduction that the paper's scope is focused on reconstruction-based SSL and does not cover other popular SSL paradigms.
>
> **Response:** Thank you for your feedback. We have modified the introduction and more explicitly emphasized this point.
>
>
> 2. > The revised manuscript presents several important findings in Section “4.4 Additional Results". For instance, the comparison involving mixup augmentation and the investigation into the model's robustness to the loss-weighting hyper-parameter α are crucial for validating the method. Placing these results in a separate section (and “only” for CIFAR-10) may understate their importance. Also, regarding the loss weighting, the robustness to  (static or continuous) raises the question whether the cosine schedule for  would be a preferable choice as it would remove the need for hyper-parameter  altogether?
>
> **Response:** Thank you for your suggestions. We have restructured the paper to better reflect the importance of these results. Specifically, we moved several key findings to the main results section (Section 4.2) and added additional experiments on CIFAR-100 (e.g., Table 5), which lead to the same conclusions. To make room for these additions, some parameter study results have been moved to Appendix A.6.
>
> Regarding loss weighting, we agree that using a cosine or linear schedule would remove the need to tune$ \alpha$. However, a well-tuned static \alpha can yield slightly better performance (as shown in Table 6), and most of our reported results are based on this setting.
>
>
> 3. > I came to share the concern raised by Reviewer Dw8m regarding the relatively low accuracies obtained by the "SL" and "SSL+SL" baselines. The authors added a footnote stating that they confirmed the convergence of the training loss. While helpful, this might point to other factors limiting generalization on the test set, and the convergence statement could be strengthened by providing the loss and validation curve in the appendix. I am unsure about the ViT-tiny architecture as this architecture type might require a lot of data and training to overcome the missing inductive bias of convolutional based architectures. For the ResNet18, I would be interested to know whether the first convolutional layer was adjusted to account for the lower resolution of CIFAR-like datasets (switching from 7x7 to a 3x3 kernel and removing the strides)?
>
> **Response:** Thank you for raising this point. Upon revisiting our ResNet implementation, we realized that we inadvertently retained the default 7×7 convolution in the first layer, rather than switching to the CIFAR-standard 3×3 kernel with stride 1. This oversight affects the ResNet results reported in Tables 2 and 4. Importantly, the majority of results in the paper—as well as all additional results provided during rebuttal—are ViT-based and therefore unaffected.
>
> After identifying this issue, we re-ran the ResNet experiments using the correct 3×3 kernel and stride 1 setting and observed the following. With the corrected architecture, MixTraining continues to consistently improve upon the SSL+SL baseline. However, in regimes where SSL+SL (with an added SSL stage) itself degrades performance relative to a strong SL baseline, MixTraining does not fully recover the SL performance. In contrast, in settings where SSL+SL improves over SL (e.g., under limited data), MixTraining further improves upon SSL+SL.
>
> These findings refine—but do not contradict—our main claims. Specifically, MixTraining consistently improves upon SSL+SL and provides a favorable compute-performance trade-off. However, when the SL baseline is already highly optimized and the addition of SSL degrades performance, MixTraining (which also incorporates SSL) may not surpass SL.
>
> We aim to be fully transparent about this implementation oversight. To this end, we have added a dedicated section (Section 4.4) summarizing the corrected ResNet experiments with 3×3 kernels and the corresponding observations. For completeness, we have temporarily retained the original ResNet results with 7×7 kernels (Tables 2 and 4) for reference, and we plan to either remove them or move them to the appendix. We would welcome the reviewer’s guidance on the preferred presentation.

---

> > ### Comment · Reviewer_Coby · 2026-02-12
> >
> > Kudos to the authors for openly discussing the implementation discrepancy regarding ResNet. And thanks for addressing my concerns and updating the manuscript again.
> >
> > I will summarize the claim the authors make in the paper as: Providing an intermediate training stage (MixTraining) between SSL and supervised-learning is beneficial in terms of accuracy and computational overhead. When replacing SSL with reconstruction-based SSL or MAE, then I see this claim to be empirically supported in the settings that are explored in the manuscript, however smaller scale. Also, I think the findings are an interesting pointer for the community.
> >
> > I will say, I find it unfortunate that the authors did not respond to reviewer Dw8m who brought up multiple comments form a practitioners perspective. (Or at least, that I cannot see the response.)
> >
> > And, there are still requests for revision that I would like to see in the final version of the paper:
> > I am still not satisfied with the position of the paper: the paper needs to be position clearly as studying the reconstruction based SSL (or even MAE) + SL setup. This is where its claims hold and it is interesting enough. While it could be that the MixTraining also has benefits for other SSL + SL setups (e.g. DINO objectives) it is not explored in this work.
> >
> > Regarding the ResNet experiments, while I do appreciate the transparency, I can imagine the somewhat mixed results as presented currently to be confusing to the reader. Because the new results don’t take away from the narrative of the paper, they can also be part of the main table. Or, I could also imagine doubling down on the MAE focus and move the ResNet experiments to the appendix entirely with the motivation that auto-encoding is not a standard for pre-training CNN architectures.
> >
> > The new paragraph that was added to the conclusion reads more like a future work section to me.
> >
> > The augmentations for TinyImageNet are not detailed in the appendix.

---

> > > ### Author Response · Authors · 2026-02-14
> > >
> > > Dear Reviewer Coby,
> > >
> > > Thank you very much for your thoughtful follow-up comments.
> > >
> > > First, we would like to clarify the response visibility. We responded to both your review and Reviewer Dw8m’s review on 11/11/2025. At that time, we were informed by the AE that there would be a delay in receiving the third review, and we did not want to leave your reviews unaddressed. Due to the OpenReview workflow prior to the official rebuttal period, author responses to early comments are only visible to the corresponding reviewers. We have now updated the responses to ensure they are visible to everyone.
> > >
> > > Regarding the ResNet experiments, we initially retained the results obtained with the 7x7 kernel for transparency. In the revised manuscript, we have removed those non-standard results and now report only the corrected experiments using the CIFAR-standard 3x3 kernel. We summarize the corrected results in Section 4.4 and clarify when MixTraining is most beneficial. We also emphasize in our paper that the majority of our experiments are conducted on the ViT architecture.
> > >
> > > Following your suggestion, we have added a dedicated paragraph in the Introduction to clarify the positioning of the paper, explicitly framing our study around the reconstruction-based SSL.
> > >
> > > We agree that the newly added paragraph in the conclusion is forward-looking. To improve clarity, we have renamed the section to “Conclusion and Future Work.”
> > >
> > > Finally, we have added the detailed data augmentation pipeline for TinyImageNet in the Appendix.
> > >
> > > We thank you again for the constructive feedback, which has helped improve the clarity and positioning of the paper.

---

### Review · Reviewer_Dw8m · 2025-10-13

**Summary Of Contributions:**

The paper proposes a training method where semi-supervised learning and supervised learning is done jointly for faster convergence and better accuracy.

**Audience:**

Yes

**Audience Explanation:**

Any training improvements are always welcome to the community.

**Broader Impact Concerns:**

No issues.

**Claims And Evidence:**

No

**Claims Explanation:**

The forward/backward pass claims are informal, and not substantiated with empirical evaluations.

The results lack standard deviations. Not all results are converged. The results show weak performance that is not typical of these networks and datasets.

The results only show simple image benchmarks of CIFAR and TinyImageNet. This is sufficient to describe the method in these simple image regimes, but does not tell us about more realistic images or datasets.

**Requested Changes:**

- The intuition of fig 3 is sensible, but how much memory, runtime, or utilisation do the different passes actually take in real-world hardware and software (ie. jax/torch)? It’s not obvious that GPU’s in real world behaves as nicely as depicted in fig3. This should be studied empirically.
- Furthermore, “merging” the two passes into one is described quite loosely. What does this merging actually do, and what properties it has? Can you formalise this more concretely?
- Alg 1 shows that the method has three phases. Is this necessary? Why not have a single mix-phase that linearly interpolates between the SL/SSL modes? Did you consider using a cosine schedule for this?
- The results need standard deviations over repeated training runs
- The results show oddly weak results. One should easily get around 95% on CIFAR10 and around 80% on CIFAR100 using common ResNet architectures. The results need to achieve realistic, competetive levels to be meaningful.
- Not all results seem to have converged. All results should show converged runs.
- I understand that running larger experiments are costly. However, full ImageNet results have been a standard part of deep learning benchmarks for over a decade, and without them it’s difficult to assess how well the method performs in more realistic data regimes or in more realistic image sizes. On the other hand, if the paper claims to only operate in low data regimes, it should also then explore some more realistic ones that CIFAR and TIN.
- In the mixup setting we interpolate between a labelled image (eg. of a cat) and unlabelled image (eg. of a dog). The prediction task is oddly to only predict a cat, and NOT to predict dog (since it was unlabelled). It then seems that the mix-phase trains only a partially meaningful objective; and the final SL phase needs to then fix this issue. Can you comment on this?
- In modern DL we would not train from scratch, but instead start from pretrained CIFAR or ImageNet checkpoints. It would strenghten the paper to include this type of finetuning experiments.

---

> ### Author Response · Authors · 2025-11-11
>
> Dear Reviewer Dw8m,
>
> We greatly appreciate your thoughtful feedback and constructive suggestions. Below, we provide detailed responses and clarifications. We have uploaded a revised manuscript with all changes highlighted in red. Please let us know if further clarification is needed.
>
> **Responses to Requested Changes:**
>
> 1. > The intuition of fig 3 is sensible, but how much memory, runtime, or utilisation do the different passes actually take in real-world hardware and software (ie. jax/torch)? It’s not obvious that GPU’s in real world behaves as nicely as depicted in fig3. This should be studied empirically.
>
> **Response:** Thank you for your comments. Figure 3 is intended to illustrate the intuitive, relative computational costs of each training phase. To further address this concern, we conducted additional experiments to empirically measure runtime, memory usage, and utilization for the SL, SSL, and MixTraining passes. The results, summarized in the table below, confirm the intuition presented in Figure 3: merging the SL and SSL passes into a single MixTraining pass reduces total runtime (as 0.248 + 0.295 > 0.317). We have added a discussion of this result in the revised manuscript.
>
> | **Metric**                     | **SL**    | **SSL**   | **MixTraining** |
> |-----------------------------|:------:|:-----:|:------------:|
> | runtime         | 0.248 | 0.295|    0.317        |
> | max_mem_mb           | 9728.42 | 22193.82 | 22237.57       |
> | cumulative_util_gpu_percent | 84.75    | 85.80    |   86.60        |
>
> 2. > Furthermore, “merging” the two passes into one is described quite loosely. What does this merging actually do, and what properties it has? Can you formalise this more concretely?
>
> **Response:** Thank you for your question. Conceptually, mixtraining merges learning signals from SSL and SL in both the data and objective spaces, producing a smooth transition rather than an abrupt switch between training phases. Below we provide a more concrete formalization, as also described in Section 3.2.2.
>
> **Data-level mixing:** We define the mixed dataset as $D_{mix} = g(D_{ssl}, D_{sl})$, where $g$ denotes a randomized mixing function. We consider two cases:
> (1) when SSL and SL share the same input data, no mixing is required;
> (2) when SSL and SL use different inputs, we construct mixed samples as
> $$
> D_{mix} = g(D_{ssl}, D_{sl})
> = ( (x_{mix}, y_{sl}) : x_{mix} = \lambda x_{sl} + (1 - \lambda)x_{ssl},
> (x_{sl}, y_{sl}) \in D_{sl}, x_{ssl} \in D_{ssl} ),
> $$
> where, for each labeled sample $(x_{sl}, y_{sl}) \in D_{sl}$, a random unlabeled sample $x_{ssl}$ is drawn from $D_{ssl}$, and $\lambda \in [0,1]$ controls the interpolation ratio. We set $\lambda = 0.5$ in our experiments to balance contributions from both datasets.
>
> **Objective-level mixing:** Using the mixed dataset $D_{mix}$, the mixtraining phase optimizes a weighted combination of SSL and SL objectives:
> $L_{mix} = \mathbb{E} [ \alpha  \ell_{ssl}(x; \theta) + (1 - \alpha) \ell_{sl}(f(x), y; \theta) ], $
> where the expectation is taken wrt ${(x, y) \sim D_{mix}}$, and $\alpha \in (0, 1)$ is a loss-ratio hyperparameter that balances representation learning (the SSL objective) and supervised learning (the SL objective).
>
> Together, these two forms of mixing create a smooth transition between SSL and SL phases, yielding both accuracy improvements (as shown in our main experiments) and computational efficiency gains (see Fig. 3 and latency results).
>
> 3. > Alg 1 shows that the method has three phases. Is this necessary? Why not have a single mix-phase that linearly interpolates between the SL/SSL modes? Did you consider using a cosine schedule for this?
>
> **Response:** Thank you for the insightful comments. In fact, our MixTraining framework can be naturally extended to incorporate a continuously annealed loss weight $\alpha$. We conducted additional experiments on CIFAR-10 (with ViT-T) using both linear and cosine annealing schedules. As shown in the table below, both variants achieve comparable performance to our original MixTraining approach. This indicates that the key advantage lies in the existence of the mixtraining phase itself, rather than in the specific scheduling of $\alpha$. We have added a discussion of this result in the revised manuscript.
>
> | **Method** | **10% Accuracy↑** | **25% Accuracy↑** | **50% Accuracy↑** | **75% Accuracy↑** | **100% Accuracy↑** |
> |-------------|:----------------:|:-----------------:|:-----------------:|:-----------------:|:------------------:|
> | **MixTraining (Original)** | 60.45% | 72.61% | 79.95% | 83.97% | 87.13% |
> | **MixTraining (Linear)**   | 61.49% | 72.21% | 79.07% | 84.82% | 86.90% |
> | **MixTraining (Cosine)**   | 60.65% | 72.03% | 79.16% | 83.40% | 87.00% |

---

> ### Author Response · Authors · 2025-11-11
> **Rebuttal Continued**
>
> 4. > The results need standard deviations over repeated training runs.
>
> **Response:** Thank you for the comment. Results reported in our paper are averaged over 4 runs with different random seeds. Below, we additionally provide results with standard deviations on CIFAR-10 (with ViT-T). As shown in the table, the standard deviations are small, indicating stable performance across runs and confirming that our main conclusions remain unaffected.
>
> | **Baselines** | **10% Accuracy↑** | **25% Accuracy↑** | **50% Accuracy↑** | **75% Accuracy↑** | **100% Accuracy↑** |
> |----------------|:----------------:|:-----------------:|:-----------------:|:-----------------:|:------------------:|
> | **SL** | 53.40% $\pm$ 0.158 | 66.03% $\pm$ 0.261 | 75.00% $\pm$ 0.432 | 79.22% $\pm$ 0.139 | 81.52% $\pm$ 0.399 |
> | **SSL+SL** | 56.79% $\pm$ 0.265 | 67.95% $\pm$ 0.155 | 77.06% $\pm$ 0.275 | 82.11% $\pm$ 0.471 | 84.69% $\pm$ 0.587 |
> | **MixTraining** | **60.45% $\pm$ 0.162** | **72.61% $\pm$ 0.357** | **79.95% $\pm$ 0.121** | **83.97% $\pm$ 0.552** | **87.13% $\pm$ 0.673** |
>
> 5. > The results show oddly weak results. One should easily get around 95% on CIFAR10 and around 80% on CIFAR100 using common ResNet architectures. The results need to achieve realistic, competitive levels to be meaningful.
>
> **Response:** Thank you for your feedback. We acknowledge that higher absolute accuracies can be achieved on CIFAR-10 and CIFAR-100 with more advanced data augmentation strategies. However, our goal in this work is not to maximize benchmark accuracy, but to enable a controlled and reproducible comparison across different training pipelines (SL, SSL+SL, and MixTraining). To ensure fairness and reproducibility, we adopt a simple set of augmentations following the practices of He et al. (2016) and Charisoudis et al. (2023), as detailed in Appendix A.3.
>
> To further address your concern, we conducted additional experiments by applying mixup augmentation to the SL and SSL+SL baselines on CIFAR-10 (with ViT-T). While mixup slightly improves the baseline results, our proposed MixTraining method consistently achieves the best performance across all data levels. This demonstrates that the observed improvements are not due to specific augmentations, but stem from the inherent advantages of our unified MixTraining pipeline, thus reinforcing our main conclusions. We have added a discussion of this result in the revised manuscript.
>
> | **Baselines** | **10% Accuracy↑** | **25% Accuracy↑** | **50% Accuracy↑** | **75% Accuracy↑** | **100% Accuracy↑** |
> |----------------|:----------------:|:-----------------:|:-----------------:|:-----------------:|:------------------:|
> | **SL w/ mixup** | 56.45% | 68.41% | 76.18% | 80.91% | 83.87% |
> | **SSL+SL w/ mixup** | 57.92% | 70.08% | 77.50% | 82.83% | 85.81% |
> | **MixTraining** | **60.45%** | **72.61%** | **79.95%** | **83.97%** | **87.13%** |
>
> 6. > Not all results seem to have converged. All results should show converged runs.
>
> **Response:** Thank you for your comments. We would like to clarify that in our main experiments where the training epochs are set to $e_{ssl} = e_{sl} = 100$, the validation loss for all methods has converged. Some validation losses did not fully converge when we reduced the training epochs to 50; however, this was intentional, as we aim to study compute-performance trade-offs under compute-constrained settings. We have clarified this point in the revised manuscript.

---

> > ### Comment · Reviewer_Dw8m · 2026-02-27
> >
> > Thanks for the comments. I'm overall happy with the responses, but the poor performance of the models needs to be fixed. This is critical for two reasons:
> > - A method has no interest if simple baselines are better
> > - This is often indicative of bugs or implementation issues, which turns the empirical evidence unreliable
> >
> > Quick googling gives some examples:
> > - https://github.com/kuangliu/pytorch-cifar reports CIFAR-10 errors of 5% using presnet18 (yours is 18%).
> > - https://github.com/weiaicunzai/pytorch-cifar100 reports CIFAR-100 errors in 20-25% range (yours is 45%).
> >
> > The baseline performance needs to be competetive, eg. match above.

---

> > > ### Author Response · Authors · 2026-02-27
> > >
> > > Dear Reviewer Dw8m,
> > >
> > > Thank you for your comments. We have already addressed the baseline performance issue and updated the manuscript on 02/10/2026. The revised results are summarized in Table 8, with further analysis in Section 4.4. Please also see our discussion with Reviewer Coby and the global clarification posted on 02/10/2026 for additional context.
> > >
> > > In Table 8 of the revised manuscript, the ResNet-18 SL baseline achieves 7.95% error on CIFAR-10 and 22.13% error on CIFAR-100, which are consistent with the ResNet-18 results reported in the repos you referenced. The ~5% CIFAR-10 error you cited is obtained with PreActResNet-18 rather than ResNet-18; in our experiments, we use ResNet-18.

---

> ### Author Response · Authors · 2025-11-11
> **Rebuttal Continued**
>
> 7. > I understand that running larger experiments are costly. However, full ImageNet results have been a standard part of deep learning benchmarks for over a decade, and without them it’s difficult to assess how well the method performs in more realistic data regimes or in more realistic image sizes. On the other hand, if the paper claims to only operate in low data regimes, it should also then explore some more realistic ones that CIFAR and TIN.
>
> **Response:** Thank you for your comments. We fully agree that large-scale ImageNet experiments provide an important benchmark. However, our study not only involves supervised learning, but also pretraining (due to the SSL part), which is computationally intensive. Given our limited compute resources, we focused on smaller-scale datasets to enable comprehensive evaluations and ablations. Nonetheless, our experiments across multiple benchmarks consistently demonstrate the effectiveness of MixTraining.
>
> To further address this concern, we conducted additional experiments under a more realistic and challenging setting by introducing random label noise into the CIFAR-10 dataset (with ViT-T). The results show that our MixTraining framework continues to achieve the best performance, even with noisy labels. We have added a discussion of this result in the revised manuscript.
>
> | **Methods** | **10% Accuracy↑** | **25% Accuracy↑** | **50% Accuracy↑** | **75% Accuracy↑** | **100% Accuracy↑** |
> |--------------|:----------------:|:-----------------:|:-----------------:|:-----------------:|:------------------:|
> | **SL** | 55.80% | 64.64% | 72.96% | 78.33% | 81.27% |
> | **SSL+SL** | 56.61% | 68.45% | 72.94% | 80.98% | 84.17% |
> | **MixTraining** | **61.18%** | **70.44%** | **75.97%** | **84.01%** | **86.13%** |
>
> 8. > In the mixup setting we interpolate between a labelled image (eg. of a cat) and unlabelled image (eg. of a dog). The prediction task is oddly to only predict a cat, and NOT to predict dog (since it was unlabelled). It then seems that the mix-phase trains only a partially meaningful objective; and the final SL phase needs to then fix this issue. Can you comment on this?
>
> **Response:** Thank you for your comments. We would like to clarify that although the unlabeled data are not directly predicted in the MixTraining phase, they are utilized through the self-supervised reconstruction loss. Consequently, information from the unlabeled samples is already incorporated into the learning signal. Empirically, we find that our approach can significantly boost model performance.
>
> 9. > In modern DL we would not train from scratch, but instead start from pretrained CIFAR or ImageNet checkpoints. It would strenghten the paper to include this type of finetuning experiments.
>
> **Response:** Thank you for your comments. We first clarify that, as stated in the introduction, our work focuses on scenarios where practitioners need to train models from scratch due to data privacy constraints or domain-specific requirements—settings that are both practical and important.
>
> That said, we believe MixTraining can also be applied in larger-scale settings with pretrained checkpoints when viewed as a **mid-training strategy**, rather than as a replacement for fully task-agnostic pretraining. As an initial validation of this idea, we conduct an additional experiment where a ViT-T model pretrained on TinyImageNet is adapted to CIFAR-10. Compared to directly fine-tuning the pretrained model (84.48% accuracy), applying MixTraining on the target dataset (as pretraining data is typically unavailable in this case) improves performance to 88.75%, indicating that MixTraining can preserve general representations while enhancing task-specific adaptation.
>
> We believe this mid-training perspective opens an interesting direction for future work, and we have added a dedicated discussion paragraph in Section 5 to clarify how MixTraining can be integrated into larger-scale training pipelines with pretrained checkpoints.

---

### Review · Reviewer_YG8C · 2026-01-14

**Summary Of Contributions:**

This paper introduces a novel training strategy that combines SSL and SL objectives by introducing an intermediate *mix training* phase that is done in between the SSL and SL phases. During mix training, SSL and SL are treated as two different tasks, and the backbone is trained in a "Multi-task Learning manner" (i.e. a shared backbone and an independent head for each "task" -- i.e. the SSL and SL objectives), additionally they propose a "mix-up" fusion of data in case the SSL and SL data distributions are different.

The work also presents several experiments that show that MixTraining achieves better performance than sequential SSL + SL training, albeit in small-scale networks and datasets.

**Additional Comments:**

No additional comments

**Audience:**

Yes

**Audience Explanation:**

I think the idea of combining the SSL and SL objectives in a multi-task manner seems reasonable and can be of interest to the community. For starters, this work can be relevant to the "efficient-training" community, where limited data is available. However, of probably more relevance, I would be interested in seeing how this MixTraining strategy can be:
 1) Scaled to larger settings, since to my understanding there is no limitation to use this in larger networks/datasets besides budget.
 2) Combined with already SSL pre-trained models (e.g. DINOv3) and use MixTraining during downstream fine-tuning to further boost performance.

**Broader Impact Concerns:**

No concerns

**Claims And Evidence:**

Yes

**Claims Explanation:**

Claims (i.e. MixTraining achieves better pareto-optimal performance than SL or SSL + SL strategies) are well supported by experiments in section 4, where the different strategies are compared for CIFAR10, CIFAR100 and Tiny Imagenet with ViT-Tiny and ResNet18.

Although the evaluation is small-scale (both in terms of network and dataset size) for the standards in SSL pretraining works, this is acknowledged and discussed by the authors, so within this scope I would say claims are supported.

**Requested Changes:**

I propose two adjustments that are not critical for acceptance yet in my opinion would strengthen the work:

1. Although claims are overall supported and ablations extensive, I missed one ablation on the mixed-dataset strategy (i.e. when SSL and SL datasets are different). I could not find any ablation comparing the mix-up strategy with any other strategy.

For instance, the simpler baseline of using only the SL data for both SL and SSL (instead of mixing them), or mixing them with other strategies (e.g. dividing the batch between SL and SSL data and then splitting the features and sending them to the relevant heads).

2. Given that the authors have not performed experiments on larger-scale settings due to computational constraints, I would encourage the addition of an additional section to discuss how the authors think the MixTraining strategy could be leveraged beyond the scope of constrained budget.

For instance, do they consider the MixTraining strategy is suitable for large-scale training as well? One potential drawback I can see is that, when training at scale, the strategy is usually  to spend a lot of compute to train a single large "foundation" model whose features are extremely versatile (i.e. useful even with frozen backbone for many downstream applications). In that sense, one would try to make the pre-training as task-agnostic as possible to keep the generalization of features and introducing MixTraining at this phase might be harmful. However, when fine-tunining a general backbone like DINOv3 for specific application, I can see MixTraining as a way to try to keep the generality of the encoder while fine-tuning it for a specific task.

3. Following-up on point 2. It might be interesting to leverage an already pre-trained model with SSL (at large scale) e.g. DINOv2-small or Dinov2-Base if budget allows and apply MixTraining as part of a fine-tuning strategy.

---

> ### Author Response · Authors · 2026-02-11
>
> Dear Reviewer YG8C,
>
> We greatly appreciate your thoughtful feedback and constructive suggestions. Below, we provide detailed responses and clarifications. A revised manuscript has also been uploaded. Please let us know if further clarification is needed.
>
> 1. > Although claims are overall supported and ablations extensive, I missed one ablation on the mixed-dataset strategy (i.e. when SSL and SL datasets are different). I could not find any ablation comparing the mix-up strategy with any other strategy. For instance, the simpler baseline of using only the SL data for both SL and SSL (instead of mixing them), or mixing them with other strategies (e.g. dividing the batch between SL and SSL data and then splitting the features and sending them to the relevant heads).
>
> **Response:** Thank you for your comments. We first clarify that we have compared MixTraining against the standard SL and SSL+SL baselines under the mixed-dataset setting, with results summarized in Section 4.2.2.
>
> The baseline that uses only SL data for both SSL and SL stages corresponds to SSL+SL results in Section 4.2.1, where SSL and SL are performed on the same dataset. As shown there, this strategy is consistently outperformed by MixTraining in the mixed-dataset setting reported in Section 4.2.2.
>
> To further address your concern, we additionally conducted an ablation that directly matches your suggested alternative: instead of mixing data during the mixtraining phase, we divide each batch into SSL and SL subsets and route them to their corresponding heads. Using a ViT-T backbone with TinyImageNet for SSL and CIFAR-10 for SL, this baseline achieves 88.51% accuracy. In contrast, MixTraining attains 89.19%, demonstrating an advantage of our proposed strategy.
>
>
> 2. > Given that the authors have not performed experiments on larger-scale settings due to computational constraints, I would encourage the addition of an additional section to discuss how the authors think the MixTraining strategy could be leveraged beyond the scope of constrained budget. For instance, do they consider the MixTraining strategy is suitable for large-scale training as well? One potential drawback I can see is that, when training at scale, the strategy is usually to spend a lot of compute to train a single large "foundation" model whose features are extremely versatile (i.e. useful even with frozen backbone for many downstream applications). In that sense, one would try to make the pre-training as task-agnostic as possible to keep the generalization of features and introducing MixTraining at this phase might be harmful. However, when fine-tunining a general backbone like DINOv3 for specific application, I can see MixTraining as a way to try to keep the generality of the encoder while fine-tuning it for a specific task.
>
> **Response:** Thank you for your comments. We first clarify that, as stated in the introduction, our work focuses on scenarios where practitioners need to train models from scratch due to data privacy constraints or domain-specific requirements—settings that are both practical and important.
>
> That said, we believe MixTraining can also be leveraged in larger-scale settings when positioned as a **mid-training strategy**, rather than as a replacement for fully task-agnostic pretraining. We agree with the reviewer that large-scale foundation models are typically trained to maximize generality, and introducing task-specific signals too early in pretraining may indeed be harmful. Instead, MixTraining is better suited to the stage where a general backbone is adapted to a target domain or task.
>
> As an initial validation of this idea, we conduct an additional experiment where a ViT-T model pretrained on TinyImageNet is adapted to CIFAR-10. Compared to directly fine-tuning the pretrained model (84.48% accuracy), applying MixTraining on the target dataset (as pretraining data is typically unavailable in this case) improves performance to 88.75%, indicating that MixTraining can preserve general representations while enhancing task-specific adaptation.
>
> We believe this mid-training perspective opens an interesting direction for future work, and we have added a dedicated discussion paragraph in Section 5 to clarify how MixTraining can be integrated into larger-scale training pipelines.
>
>
> 3. > Following-up on point 2. It might be interesting to leverage an already pre-trained model with SSL (at large scale) e.g. DINOv2-small or Dinov2-Base if budget allows and apply MixTraining as part of a fine-tuning strategy.
>
> **Response:** Thank you for the suggestion. Please refer to our response to Point 2 above, where we discuss this setting and provide an initial experimental validation, along with a dedicated discussion in Section 5.

---

### Author Response · Authors · 2026-02-11

Dear Reviewers,

We identified an implementation issue in the ResNet experiments, where the first convolutional layer inadvertently used a 7×7 kernel instead of the CIFAR-standard 3×3 kernel with stride 1. To address this, we added new ResNet results using the corrected 3×3 kernel and included a dedicated subsection (Section 4.4) analyzing when MixTraining helps and when it does not. Importantly, the majority of results in the paper—as well as all additional results provided during rebuttal—are ViT-based and unaffected. These additions refine but do not contradict our main claims.

---

### Decision · Action_Editor_bbwE · 2026-03-19

**Recommendation:** Accept as is

**Audience:**

Yes

**Audience Explanation:**

Yes, absolutely. Based on the reviewers' feedback, there is a clear consensus that the findings of this paper will be of interest to the TMLR audience.

**Claims And Evidence:**

Yes

**Claims Explanation:**

All three reviewers recommend a "Leaning Accept" for this manuscript. The paper proposes a training procedure that mixes supervised and semi-supervised paradigms for image classification, demonstrating both accuracy and efficiency improvements. The reviewers agree that the underlying intuition is sound and that the authors' claims are properly supported by empirical evidence within the chosen experimental scope. Furthermore, the authors were proactive during the review process, identifying and correcting an implementation issue regarding ResNet kernel sizes and subsequently adding valuable analysis (Section 4.4) detailing the specific conditions under which their proposed method succeeds or fails.